# Sustainable Heritage Tourism: Native American Preservation Recommendations at Arches, Canyonlands, and Hovenweep National Parks

**Richard Stoffle [1,\*], Octavius Seowtewa [2], Cameron Kays [1] and Kathleen Van Vlack [3]**

1   School of Anthropology, University of Arizona, Tucson, AZ 85721, USA; b1lairkays@email.arizona.edu
2   Pueblo of Zuni, Zuni, NM 87327, USA; oct.seowtewa@gmail.com
3   Living Heritage Research Council, Cortez, CO 81321, USA; kvanvlack82@gmail.com
\*   Correspondence: rstoffle@email.arizona.edu; Tel.: +1-520-907-2330

**Abstract:** The sustainable use of Native American heritage places is viewed in this analysis as serving to preserve their traditional purposes and sustaining the cultural landscapes that give them heritage meaning. The research concerns the potential impacts of heritage tourism to selected Native American places at Arches National Park, Canyonlands National Park, and Hovenweep National Monument. The impacts of tourists on a heritage place must be understood as having both potential effects on the place itself and on an integrated cultural landscape. Impacts to one place potentially change other places. Their functions in a Native American landscape, and the integrity of the landscape itself. The analysis is based on 696 interviews with representatives from nine tribes and pueblos, who, in addition to defining the cultural meaning of places, officially made 349 heritage management recommendations. The U.S. National Park Service interprets Natives American resources and then brings millions of tourists to these through museums, brochures, outdoor displays, and ranger-guided tours. Native American ethnographic study participants argued that tourist education and regulation can increase the sustainability of Native American places in a park and can help protect related places beyond the park.

**Keywords:** sustainable heritage tourism; native American heritage places; United States National Parks; Arches National Park; Canyonlands National Park; Hovenweep National Park

---

## 1. Introduction

Native American heritage places exist in many U.S. national parks. Most are interpreted as archaeology ruins, but natural places like canyons, mountains, springs, streams, fields of plants, and the nesting homes of raptors are often not known as places of cultural importance to Indian people and generally are neither a part of why the park was established nor a part of its ongoing heritage management. Ethnographic Overview and Assessment (EOA) studies, funded by the National Park Service (NPS), are used to identify places of cultural importance to Native American tribes and pueblos that are a part of their cultural heritage.

This analysis is about the potential impacts of heritage tourism to selected Native American places at Arches National Park, Canyonlands National Park, and Hovenweep National Monument (Figure 1). Sustainability and heritage tourism has been a world priority since the U.N. Sustainable Development Goals were adopted [1–3]. Since then, sustainable development has been a guiding principle for the management of tangible, intangible, and natural heritage [4]. Sustainable development is viewed by UNCED 1992 as proceeding with the paradigm that considers environmental, economic, and social goals [2].

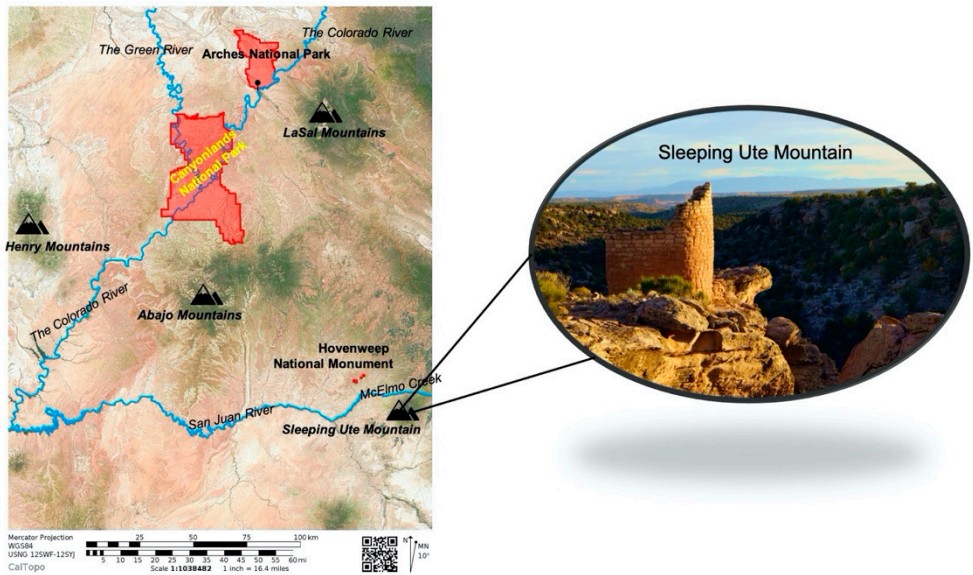

**Figure 1.** Region Map (Map Courtesy of Stoffle, Van Vlack, and Kays).

United National sustainable goals and guidelines are looked to by many countries. As a result, there is an extensive research literature on sustainable heritage tourism [5]. Many of these studies focus on urban heritage, which is viewed as the most common sitting of the world's cultural heritage [6]. The body of literature has matured, now having indicators organized in models and applied as comparisons withing nations and across regions [7,8]. There are even studies of tourism and the spirituality of place in Nepal [9]. Findings are readily used in agency and state planning.

For U.S. national parks, however, the UN guidelines are not applied. Instead guidance derives from the National Park Service, Management Policies [10] which are directly tied to statutes and regulations. Our research team conduced the last large-scale tourism survey involving 1796 site-intercept interviews in four national parks [11]. Soon afterwards, the NPS placed all visiting tourists off limits for study. Today there are few surveys of tourists in national parks, and no known surveys that involve tourists and native heritage issues. There are no indicators of sustainability or models for any resource, especially cultural heritage. The current analysis is uniquely about tourism in three extremely rural parks and from a Native American heritage perspective.

## 2. Operationalizing Concepts

The analysis uses a number of generally understood heritage concepts that are subsequentially operationalized for this Native American study. These concepts are (1) heritage places, (2) heritage landscapes, (3) study findings, (4) study recommendations, and (5) heritage sustainability. In the U.S. Native American heritage places are defined by federally recognized tribal and pueblo governments through their official cultural representatives. Generally, this process is governed by National Register Bulletin #38 regarding Traditional Cultural Properties (TCP) which specifies that only a cultural (ethnic) group can define a TCP [12]. Cultural landscapes are similarly defined in a National Register Bulletin #30 [13] which defines Native American landscapes as a kind of rural cultural landscape. Like native heritage places. Only a cultural group can identify heritage landscapes.

Ethnographic study findings, such as those presented here, therefore define cultural landscapes with data derive from systematically listening to and recording statements made by representatives of cultural groups who officially appoint tribal members to represent cultural understandings about places and landscapes. Tribal representatives have experience with how tourists impact traditional heritage places, so their assessments are grounded in dozens to hundreds of similar ethnographic studies conducted in their aboriginal lands. Mr. Seowtewa, a co-author of this paper, is the head of a cultural committee for the Pueblo of Zuni who has over more than four decades been engaged in

these studies. His committee of specialists and knowledgeable cultural experts from other tribes and pueblos are the foundations of our understanding of both how tourists impact heritage areas and what the NPS can do to effectively mitigate these impacts.

Confidence in study findings increases to the extent that representatives and cultural groups agree; that is, have similar if not the same identifications. Only representatives can identify the causes of and mitigations for spiritual and physical impacts to their heritage places and landscapes. Thus, heritage sustainability under pressure from tourism is defined as the continued ability of the tribes and pueblo people to utilize places and landscapes in culturally appropriate ways.

These three NPS studies assess how tourist behaviors can impact the sustainability of both places and landscapes. It is thus first necessary to understand the cultural meanings of places and how these are integrated into heritage landscapes [14]. Stoffle, Halmo and Austin [15] defined kinds of cultural landscapes; while Toupal et al. [16] provide a methodology for landscape studies. Rossler [17] argues that cultural landscapes are a significant component in sustainability development because they are important in the economic and social life of many countries. Fard and Saboonchi [18] document the landscape as symbolic of nature in Iran.

Heritage places and landscapes can be impacted physically and spiritually by foot traffic erosion, damaging places with graffiti, collecting physical offerings left on the ground, behaving in ways that insult the places like loud talking, and conducting New Age type ceremonies that conceptually conflict with the spirituality of the place. Millions of tourists visit arches and so these areas are constantly being eroded. Off trail foot paths created by a few tourists can damage residual rock walls and lead to touching rock paintings. Offerings left after ceremonies are complex and unless understood in consultation with Native Americans can be overlooked. For example, prayers at a place can be accompanied by an offering of sage placed under a stone, crystals, red jasper, and small pieces of pottery. These objects are not readily identified as artifacts and thus typically are not regulated, so tourists move them or take them away. Loud talking and throwing stones are behaviors never allowed at native places. Spiritual places have protocols for visitation by Native Americans who approach with reverence, announce the purpose of the visit, and specify what is needed such as medicine for cures, balance to reduce conflicts, and rain so the streams flow. Certain kinds of ceremonies generally understood by the culturally associated native peoples should occur at heritage places. Other kinds of inappropriate behaviors are when New Agers conduct their own ceremonies at sacred sites and when people spread the cremation ashes of relatives in kivas. These can not only cause spiritual damage at heritage places but disrupt the integration and function of cultural landscapes.

While the notion of heritage place is comparatively easy, Native American heritage cultural landscapes are a more complex notion. In the U.S., *cultural landscapes* are recognized as historic properties that can be protected by being placed on the National Register of Historic Places by the federal government. Officially they share a geographical area that historically has been used by people, or shaped or modified by human activity, occupancy or intervention and that possess significant concentrations, linkage, or continuity of areas of land use, vegetation, buildings and structures.

Native American people tend to understand cultural landscapes as a series of heritage places linked by culturally important spaces, both having traditional purposes that were defined at Creation and subsequently used appropriately by native people [15]. These landscapes embody physical and spiritual connections and exist in multiple dimensions. Native societies form relationships with their environment that are manifested in these complex intersections among themselves, material culture, history, and geography [19]. For native people, these intersections are grounded in their epistemologies and oral traditions [20,21].

A native cultural landscape example is pictured in Figure 2. It is of hoodoos at Arches National Park in the foreground and on the horizon the La Sal Sky Island massif (i.e., spatially and ecologically isolated high mountains) as they interact spiritually with each other. This is a visually and hydrologically integrated heritage cultural landscape that extends far beyond the park. This landscape crosses the Colorado River which carries prayers downriver to the ocean.

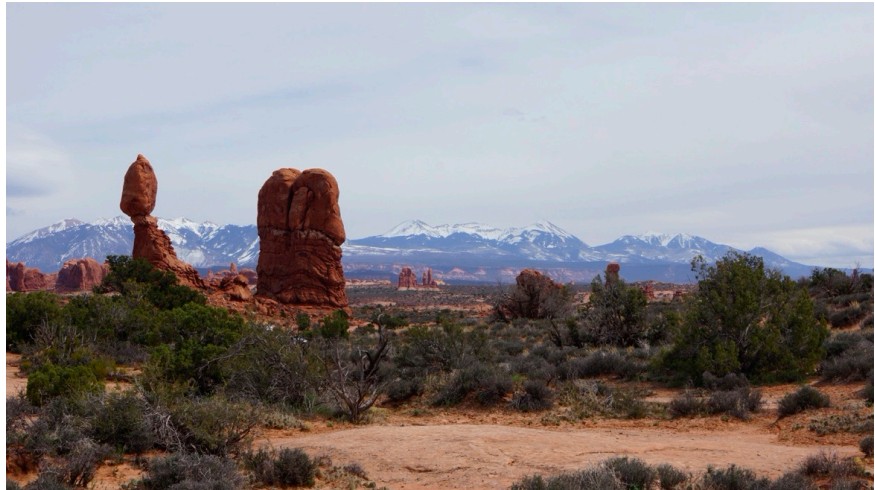

**Figure 2.** Landscape of Hoodoos and Sky Islands (Photo Courtesy of Stoffle, Van Vlack, and Kays).

Native Americans participating in these ethnographic studies stipulate that some heritage landscapes are integrated by the movement of water and involve direct relationships between high snow-covered mountain massifs, local streams, and distant springs at sacred places. Movements along these waterways occur in this and other dimensions. Waterways are traveled both ways by spiritual beings and prayers in what is perceived as a *Living World*. Some streams flow into the Colorado River, which passes through the Grand Canyon to the Gulf of California. Prayers are sent downstream to the Gulf to ask that it send rain back to the Sky Island mountains. When this occurs, prayers of thanks for the snow and rain that arrive are sent up to the mountains and they again flow downstream to the Gulf. Waterways are the *living veins of Mother Earth*; a commonly held Native American epistemological truth or veritas.

This analysis is based on the native notion that the *Earth is alive*. This is an epistemological foundation of Indian culture or what Rappaport [22] calls an *ultimate sacred postulate* and what Goldman [23] calls a philosophical *primitive*. The concept of a living universe is essential for understanding Native American heritage places and their relationships in heritage landscapes [24,25].

Cultural logic and oral history can be used to establish what Tilley [26,27] calls the *materiality of stone* and the *phenomenology of landscapes*. Tilley's work is focused on the Middle Bronze Age people of the British Isles and the Britany Coast of France who talked and interacted with massive stones called *menhirs*. They also organized ceremonial areas into line-of-sight landscapes containing culturally special topographic features. His work is a model for our thinking about places and landscapes. He argues that questions of cultural association, ceremony use, and heritage meaning can neither be proven nor disproven. He asserts that the heritage meanings of non-artifactual natural places and resources are either believed or not by contemporary cultural groups. Such places officially are called Traditional Cultural Properties, which legally only exist in the culture of culturally associated contemporary ethnic groups.

According to Native American beliefs, the universe is alive in the same way humans are alive and fully sentient. It has physically discrete components that some call *elements*, and an energy source that brings them alive that is called *Puha* in the Numic language, or something that can be translated as 'Creation energy' or 'power' [25]. Elements like mountains, rivers, fauna, flora, wind, and minerals have most of the same characteristics as humans, including the ability to communicate, to help other elements, the power to accomplish their own goals called *agency*, and even the capacity to lie. The heritage places are understood by Native American people as *living beings* through this epistemological perspective. The places remember what ceremonies were held and what was said to them. Places can share these ancient songs today. These living heritage beings are interacting with tourists as they visit these national parks.

### 3. Background

The National Park Service (NPS) was established by the U.S. Congress in 1916 through the Organic Act, which made them responsible for managing all national park lands and resources that had been set aside by Congress beginning in the late 1800s, as well as all new parklands declared from this time forward. The primary dual mandate of the new NPS was (1) to preserve natural and cultural resources within specific parks and (2) to provide public access to these resources. These dual goals can be in conflict when preservation and access have different results. When the NPS mission was updated in 2000, it contained an increased emphasis on working with partner organizations and communities such as Native American tribes and pueblos to extend the benefits of heritage conservation.

Public education has become a primary means of accomplishing both preservation and public access. The NPS today educates tens of millions of people. Each NPS park has one or more tourist visitor centers with a free interpretative video that explains the most charismatic components of the park. Museums provided further information. At the curio center, hundreds of educational materials are sold, from maps, to postcards, to children's coloring books, and science books. Beyond the visitor center are dozens of trails lined with interpretative signage and displays (Figure 3). Park rangers provide guided educational tours to interesting locations. Tourists come to parks to learn about the park, be wowed by its natural splendor and cultural interests, and cautioned about how to preserve these treasures.

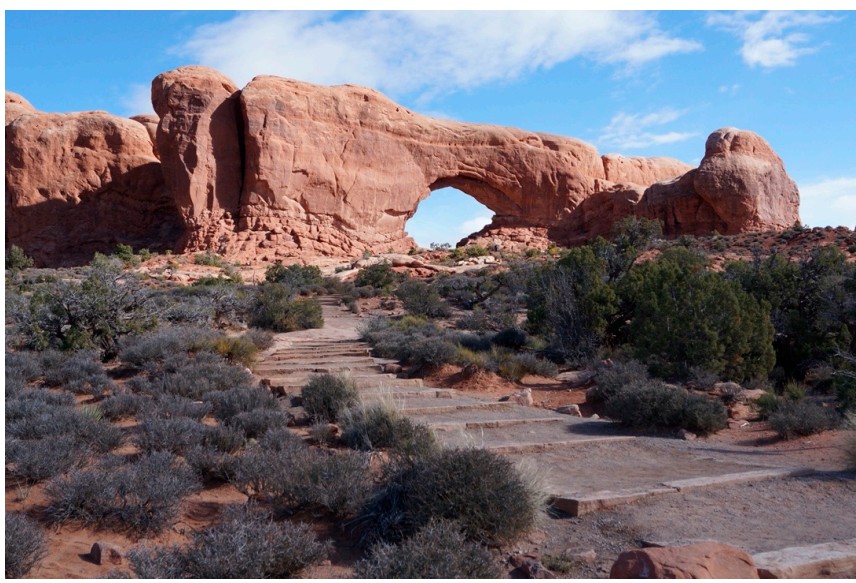

**Figure 3.** Tourist Path to Arches (Photo Courtesy of Stoffle, Van Vlack, and Kays).

Our research understandings of national park interpretations are framed by the broader issue of how have Native American peoples been publicly represented by the U.S. government. Many parks and national museums have celebrated colonial progress narratives that built on the U.S. national narrative of Manifest Destiny, where civilized settlers were destined by God to expand across North America and replace savage Native Americans [28]. In keeping with this narrative, many Indian people were removed from national parks when established and ignored when interpretative displays were developed [29–31].

This essay considers how national parks might reflect diverse, intersecting, and transforming identities and even center Native American cultures in park interpretations. American anthropologists since Boas have argued for an accurate interpretation of the culture of Native American and other indigenous peoples [32]. Native American peoples and scholars of American culture have led efforts to decolonize national interpretations and support Native Americans seeking to tell their own stories [33,34]. Critical for achieving this goal is the culturally accurate interpretation of heritage

places and the implementation of tourist education and regulations that permit the sustainable use of these places.

*Study Area*

This analysis focuses on lands and resources within three national parks in southeastern Utah and western Colorado. Park boundaries are used in management and interpretation; however, this isolates park places and tourist activities from broader Native American heritage landscapes. While parks often exclude discussions of places located outside the park, our studies document that from a Native American perspective the heritage meaning of park places cannot be fully understood without reference to broader functionally interconnected cultural heritage landscapes.

Two of the three parks are part of a much larger aboriginal cultural landscape called the *American Indian Crossing of the Colorado River* (AICC). Moab, Utah, located south of Arches and north of Canyonlands, is one of a few places used to cross the Colorado River. This occurred because the Moab area lacks the very high canyon walls, which elsewhere make getting down to and up from the Colorado River channel difficult. This landscape is understood as being bounded by three topographically large Sky Islands massifs: La Sal Mountains to the east, Abajo Mountains to the south, and the Henry Mountains to the west. The region is dissected by the Colorado River and the Green River, which unite in Canyonlands and then flow into the Grand Canyon. In addition, there are a variety of medium-sized creeks and rivers, all of which flow from Sky Island massifs into the Colorado River.

The crossing of the Colorado River is the lifeline of social interaction and ceremonial activity centered on a traditional trail that connected the entire region [35,36]. For two thousand years, Indian life at and near this trail crossing involved large residential communities primarily supported by irrigated agriculture along the rivers and streams. These people had diverse hunting and gathering areas in the surrounding mountains. Activities often involved hundreds of people participating together over months at a time. This way of life was maintained by a complex system of functionally integrated ceremonies often collectively held by religious leaders from distant communities. Native American stories told about activities in this area cannot be fully understood without reference to places located throughout the region.

Hovenweep is situated in another distinct but overlapping cultural landscape, which has been termed the Mesa Verde World [37]. This region—named because of the charismatic national park, Mesa Verde National Park—is defined by archaeologists as an integrated complex of common cultural traditions and architectural styles, spanning from the junction of the Colorado and San Juan Rivers to the flats on the southwest edges of the San Juan Mountains. This landscape has been occupied for up to 17,000 years as documented by early spear points [38–40] and as suggested by early dates from Monte Verde in Chile [41], Paisley Five Mile Caves in Oregon [42], and Sonora, Mexico [43]. The functional integration of the region, however, occurred over a span of 800 years, from approximately AD 500 to AD 1300. During this period, the weather was warmer, additional rain fell and agriculture expanded onto the fertile mesas where a significant number of farms were established over hundreds of square miles. The farms produced an agricultural surplus that supported complex cities and religious centers headed by a priesthood class. The people of the Mesa Verde World maintained a distinct material culture and religion that physically and spiritually integrated these people among themselves and others living in cultural centers hundreds of miles away.

According to participating tribal and pueblo representatives, Hovenweep is one of these Mesa Verde World ceremonial complexes. It was deliberately placed and constructed for the purpose of revitalizing the land through spiritual practice after a series of severe droughts. The world was out of balance in the AD 1200s and the religious elite from all over the region needed to work together to make it right again.

## 4. Methodology

As native peoples reconnect with aboriginal homelands, they share heritage perspectives to influence interpretation, management, and sustainable use by tourists [44,45]. The NPS conducts formal consultation on a government-to-government basis to collect new information to be used in museums, brochures, outdoor displays, and ranger-guided tours. One type of official study is the park-funded Ethnographic Overview and Assessment (EOA).

This analysis involves three EOA studies conducted by our University of Arizona research team for Arches [35], Canyonlands [36], and Hovenweep [46]. It is based on formal and informal EOA interviews with representatives of the following nine tribes and pueblos who participated in one or more of these studies: (1) Acoma Pueblo, (2) Ute Indian Tribe of the Uintah and Ouray Reservation, (3) Southern Ute Indian Tribe, (4) Paiute Indian Tribe of Utah, (5) Kaibab Band of Paiute Indians, (6) Navajo Nation, (7) Hopi Tribe, (8) Zuni Pueblo—A:shiwi, and (9) Santa Clara Pueblo. The analysis is based on 696 ethnographic interviews (168 at Arches, 316 at Canyonlands, and 212 at Hovenweep).

The NPS funded the EOAs to understand the cultural meaning and importance of the natural and archaeological resources within the parks that are culturally associated Native Americans. The studies were officially designed to be participatory [47] so the tribal and pueblo representatives shared their opinions as to where the study visits should occur and what are the most salient topics to be discussed. The studies were funded to meet certain park management and interpretation goals, including (1) learning about tribal perceptions, (2) knowing oral histories of the ancient farming communities, (3) understanding the intended purposes of the peckings and paintings, and (4) hearing about the contemporary uses of the abundant and rare plants located throughout the parks.

Our research team has worked with the Native American people for over 50 years; during this time a strong research partnership has been established. This partnership has directly influenced how we approach research projects and our research methodology, which involves the use of mixed methods [48,49] and triangulation [50]. The mixed methods approach involves collecting qualitative and quantitative data, and where there is convergence, confidence in the findings grows considerably [51].

We have developed seven survey instruments that have been developed with the assistance of official tribal representatives and the approved by participating tribal governments. Many of these instruments, such as the Site Form, Rock Art Form, Cultural Landscapes Form, Pilgrimage Connection Form (See Appendix A), have been administered since 1997. Each of these were utilized during these EOA studies.

Interviews occur with the understanding that confidentiality will be respected during the duration of study process. Protocols have been negotiated with each of the participating tribes:

- Each tribal representative is afforded the right to a private interview. No text or tape is released without the full consent of the tribal representative. Text and tapes are returned to the tribal representative at their request.
- Ethnographers extract from the private interview pertinent site interpretations and evaluations of impacts and combine these into a composite text, which can have minority opinions.
- Ethnographers send to the tribal representatives the composite text that builds on the private individual interviews but reflect some ethnographic synthesis. This text generally does not involve references that could identify the representatives who contributed to the text but being quoted is an option. Tribal representatives have the right to add, subtract, and correct the composite text.
- Tribal government reviews evaluate, based on any other criteria, and approve or disapprove the composite text. Tribes can elect to identify and recommend selected sites as Traditional Cultural Properties (or eligible for designation as Traditional Cultural Properties) and Sacred Cultural Landscapes. At this point the entity funding the research will receive the text. Once the tribes approve the text, the content cannot be altered and will be accepted as is by the NPS.

Shared cultural perspectives potentially serve to inform park interpretations in the visitor center museum and along hiking trails and provide new ideas for park management, especially how visitors

should interact and treat heritage places and landscapes. In most instances, tribal and pueblo representatives shared common perspectives, thus providing a clear and unified story about parklands, natural resources, and objects. Overall, these constitute a *Native American perspective*, though differences exist. These perspectives enrich what was already being told in park interpretations or being practiced in land management. In other cases, they bring new ideas to Native American history, the cultural meaning of the land, the cultural centrality of natural resources, and the traditional use of objects. It is important to note that after reviewing and formally commenting on park interpretations, the participating tribal and pueblo representatives were supportive of park museums and management. The representatives also agreed that the traditional people of these areas each have valid alternative voices.

## 5. Findings

The following three cases illustrate new Native American park resource identifications and recommendation for preserving heritage places. These consultation recommendations do not obligate the NPS to change or add to interpretative displays. Recently, however, these parks have demonstrated an interest in having their teachings reflect Native American voices. Today many of these recommendations are being implemented.

The phrase "contribute a voice to a discussion" is commonly understood by scholars, but here the technical concept of voice is used in the studies to frame the analysis and to assess the recommendations. Anthropologists have studied voice as a set of relationships of agency or power, that are established between the speaker and the hearer [20,52]. The concept of voice necessarily raises questions of who has authority to claim a voice for a social group or community. This authenticity issue was raised in these EOA assessments of park interpretative displays that quote a Native American without there being a clear evidence of tribal approval. The notion of voice signals the possibility of convergent and divergent perspectives both between and within communities.

The concept voice authenticity therefore raises the question of who it is that the speech represents. Clearly, in studies of the relationships between tribes, pueblos and parks, the voices incorporated in any report should represent, with some level of confidence, the ethnic groups involved. To seek the highest levels of confidence, a methodology was used in these studies that involves (1) tribes/pueblo governments agreeing to participate in the study; (2) official representatives being chosen by their tribal/pueblo government and their participation supported; (3) confidential interview situations provided by the ethnographic research team and the park; (4) all report text being first reviewed by the representatives, their cultural departments, and if required their tribal/pueblo councils; and (5) the NPS agreeing not to modify either tribal/pueblo interpretations or recommendations.

It is beyond the scope of this paper to discuss most tribal and pueblo recommendations to the NPS. Representatives made 349 heritage recommendations: 103 at Arches, 150 at Canyonlands, and 96 at Hovenweep. Table 1 presents the 349 recommendations and cross tabulates them by park and topic.

**Table 1.** Recommendations by park and type.

| Park | Signage/ Museum | Site Management | Teaching/ Visiting | Resource Collection | Total |
|---|---|---|---|---|---|
| Arches | 33 | 31 | 30 | 9 | 103 |
| Canyonlands | 78 | 47 | 19 | 6 | 150 |
| Hovenweep | 76 | 8 | 12 | 0 | 96 |
| Total | 187 | 86 | 61 | 15 | 349 |

All recommendations were approved by tribes and pueblos. In each park, recommendations broadly fell into four categories. First, the largest category involved changing signage and updating museum information at both visitor centers and interpretive sites along trails to reflect native interpretations and new definitions of traditional purposes. Second, many suggestions were made about how places could be managed to better reflect their traditional purpose and contemporary meanings to Native Americans. Third, requests were made to bring groups of tribal/pueblo youth

and other members into the park for camping, teaching, and ceremony. Fourth, many representatives requested permission to collect traditional plants and natural resources like clay and paint pigments.

## 6. Case One: Meaning of a Spring

The first heritage landscape is located in Canyonlands. It is defined by water movement that connects places in the land including a special spring that emerges from under a large sandstone formation. The case illustrates the native meanings of springs. Cave Spring is surrounded by slick rock shelfs, sandstone alcoves, and dense plant life. Tourist access to this location is by good dirt road, and well-maintained trail near the car parking. There are interpretive signs, but none convey a native perspective. The many tourists who visit have full access to the spring and associated cultural features (Figures 4 and 5).

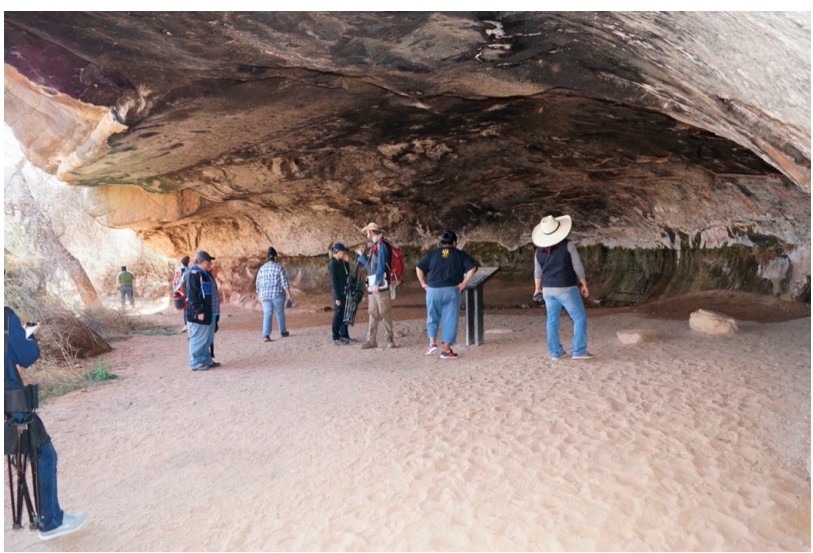

**Figure 4.** Cave Spring Rock Visitor Total Access (Photo Courtesy of Stoffle, Van Vlack, and Kays).

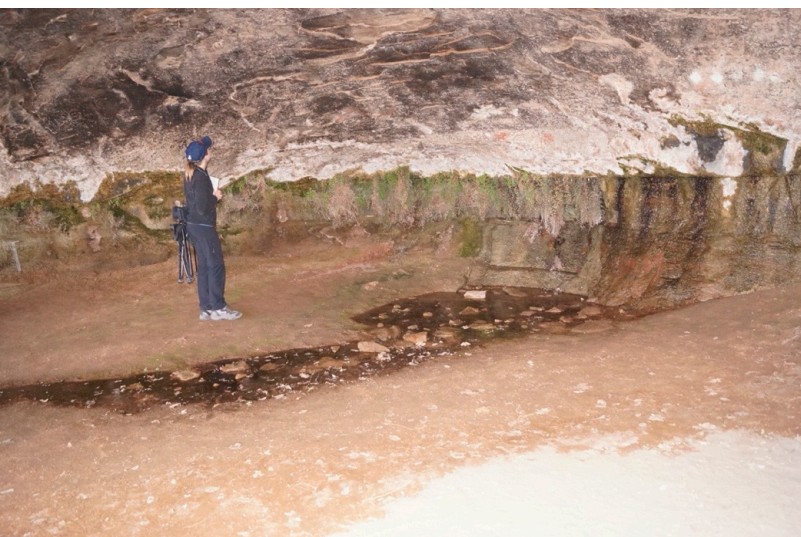

**Figure 5.** Cave Spring Before Restoration, Rocks Thrown in Spring (Photo Courtesy of Stoffle, Van Vlack, and Kays).

Water is incredibly important and sacred in native culture. One representative noted that "We take the energy from the water, like we do from plants. It has emotions, energy, memory. Water gives us strength." Cave Spring is fed by rainwater that falls in shallow depressions on top of the sandstone

cap rock, and then percolates through the porous sandstone until it reaches impermeable layers of rock. It comes to the surface again under a large overhang. The water at Cave Spring is thus especially sacred because it emerges from the ground. Ultimately, the rain and snow come from prayers sent to the surrounding Sky Island who call down the moisture. One representative observed the following:

> *This was special. The water came from underground, filtered maybe through the rocks but it came out. It is special because of this curing ability, the pureness of the water, they were coming here just for that purpose.*

Red, yellow, and white handprints, or paintings in the shapes of handprints, are located above and adjacent to the spring (Figure 6). The presence of the handprints intensified the importance of the spring because they represent proof of Native use of the spring. Even today, according to one representative, the spring is not viewed as abandoned as some archaeologists say, but rather as continuously utilized:

> *Oh, there is someone here. It is not empty. There is always a presence. We do not mean to intrude; we are here today looking around and giving acknowledgement of your being there and being present.*

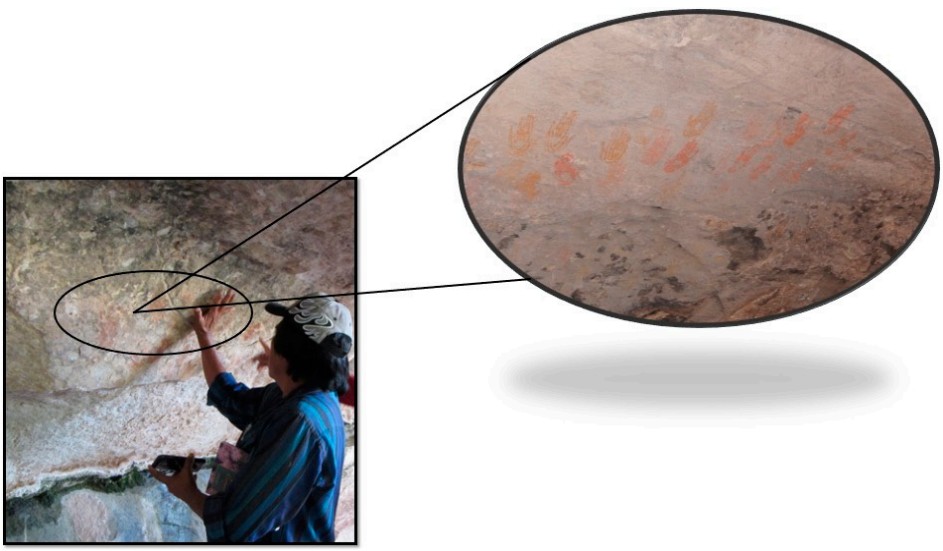

**Figure 6.** Cave Spring Paint Handprints Can be Touched by Tourists as Described by a Tribal Representative (Photo Courtesy of Stoffle, Van Vlack, and Kays).

The handprints were viewed as personal prayers and ceremonial markers. As one representative noted, "I am sure they held ceremonies here, which is why those symbols are here. That is a sacred spring." The presence of the handprints at the spring contribute to the sacredness of the spring, enhancing the sense of connectedness that Native American representatives felt with the land, the spring, and their ancestors. Tourists should not touch or damage the hand prints.

Under the overhanding rock are bedrock grinding slabs that reinforced representatives' interpretation this spring is a sacred place (Figure 7). These were for grinding seeds, plants, and berries for paint, food, used in ceremony and medicine. Grinding slabs at this location enhanced native perspectives that the spring was a central location for repeated ceremony by nearby inhabitants who had permanent irrigated farming villages along Salt Creek about a mile away; "There is a reason why the people would be coming here. Not just once, but periodically, regularly, to do ceremonies or whatever they did here, [it] was very important for them, to them, that they have grinding slabs." Tourists should not use stones to grind in the slabs.

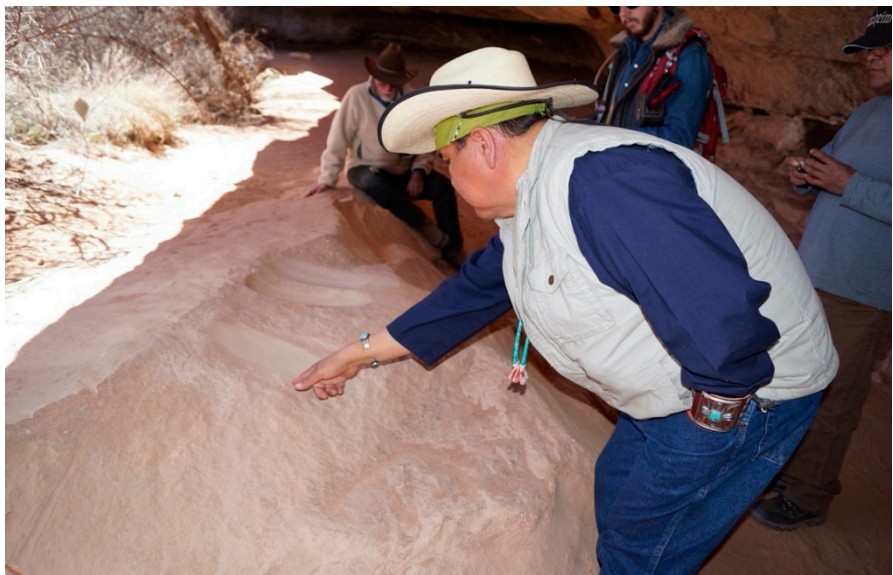

**Figure 7.** Cave Spring Grinding Slabs, Tourists Pick Up Nearby Small Rocks and Grind in Slabs (Photo Courtesy of Stoffle, Van Vlack, and Kays).

*Recommendations*

The meaning of this spring is critical for understanding the management recommendations provided by the Native American representatives. The Native people participating in these EOA studies perceive springs as portals between worlds. Springs provide the two-way transportation of power, energy, ceremonies of appreciation, and prayers for rain and snow. Springs can contain spiritual beings, like Water Babies who can become spirit helpers for rain shaman (Stoffle and Arnold 2003).

When treated in a culturally appropriate manner, springs are a key element in the balance of life. Tourists should not throw rocks into the spring, use the bedrock grinding areas, or touch the hand painting on the walls. A barrier with signs conveying an interpretation of the spring as a sacred area should be erected to keep tourists at a distance and influence their behaviors while at the spring.

The representatives recommended that they be permitted to develop an elaborate process of talking with, clearing, and protecting the springs, especially those located under overhangs. A foremost concern was to re/establish the Native American–spring relationship, which has been attenuated by the removal of Native people from the park. Although Native people continue to send prayers back to these springs (as well as to creeks and rivers), proper interaction involves on-site prayers and offerings. Second, springs need to be cleaned on a regular basis by people who have Traditional Ecological Knowledge about how to talk with them. Such cleaning involves minimal modification but some removal of plants, dirt, and rocks.

Tourists should visit springs by park roads and developed trails, however the springs require signs indicating their sacredness and requesting respect from park visitors. This is especially important for those springs that also have associated paint mineral deposits and painted handprints on the overhanging ceilings.

## 7. Case Two: Hovenweep Prayer Towers, Kivas and Springs

A second heritage area is in Hovenweep, which has multiple elaborately constructed structures located on both water features that are part of a water-based heritage landscape and line of sight visual viewscape-based heritage landscapes. These two kinds of landscapes are centered on Hovenweep. Tribal and pueblo representatives stipulated that these landscapes helped determine the location of the many structures that were constructed at Hovenweep.

The Hovenweep structures are among the most complex and carefully built in the Southwestern U.S. They are located within a few miles of each other at the heads of shallow canyons (Figure 8).

Each canyon head today has a prominent spring but in the AD 1200s when most of the structures were made the springs would have emerged at the head of the canyon and flowed down the canyon to adjoining streams, then on to the McElmo River, then to join with the Colorado River, and then to the Gulf of California. This is the downstream portion of the hydrological landscape. The upper portion of this water landscape begins to the north with the Sky Islands and high mesas. Representatives commented about these issues:

> *Look at the layout of where all these different settlements are. They are right at the [head] of the canyons, right at the beginning of where the water starts. It is not halfway down. That is the only reason that we have that, they are religious gathering spots.*

> *The aquifer, the watershed, all have connections underground. This whole world has connected water underneath. So all of that water has to come out at a specific place. Coming out at the place where structures are built, not to defend it, but to pray for it. To have that water constantly flowing.*

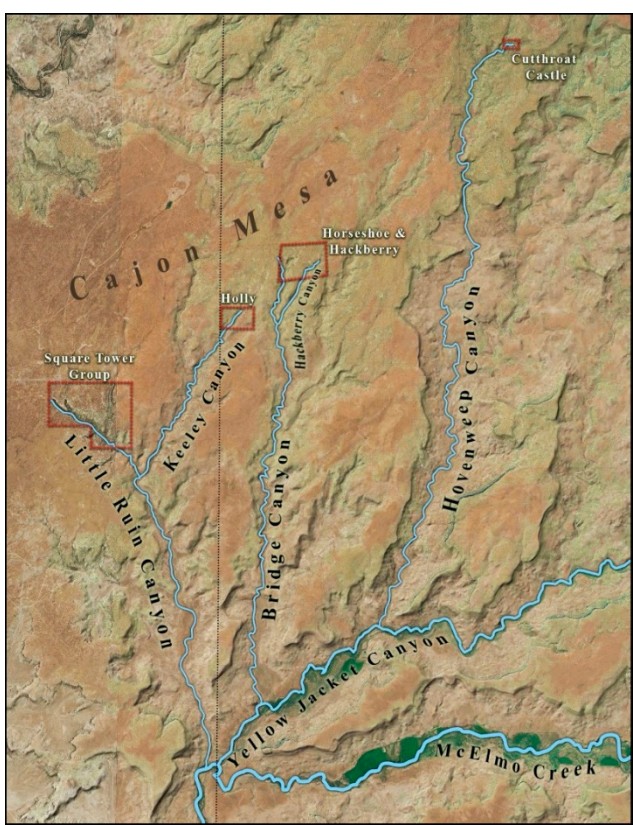

**Figure 8.** Hydrological Landscape of the Hovenweep Area (Map Courtesy of Stoffle, Van Vlack, and Kays).

The canyon rims and bottoms all have a complex of towers and kivas (Figure 9). Kivas are subterrain circular ceremonial areas that are entered through the roof. They have benches, a fire pit, a *sipapu* entrance to another dimension, and a hidden entrance to a secret tunnel to the towers on the surface. The towers are often built on top of large stones and the kivas tend to be connected at the base of the tower with a tunnel. Many additional tower-kiva structures are found in the canyon bottoms especially near the spring. The canyon rims are lined with additional structures having multiple story buildings that are often square or rectangular. Some of these too are connected with the canyon bottom kivas through holes in the canyon rim.

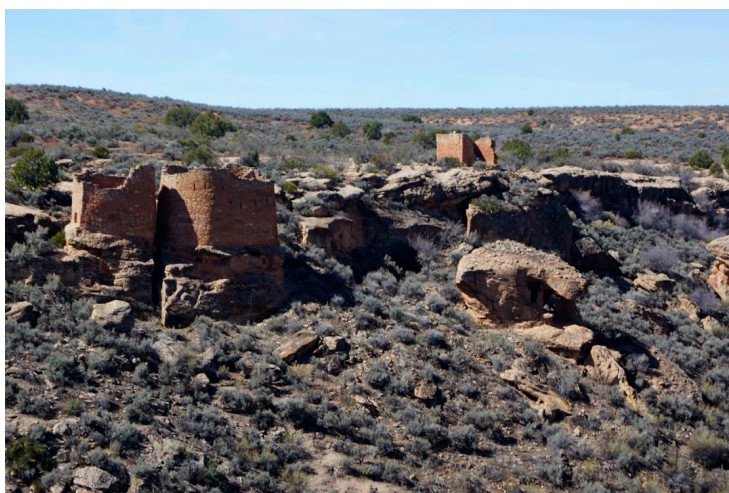

**Figure 9.** Tower/Kiva Complex, Towers on Rim, Tunnel in Crack to Kiva on Bottom (Photo Courtesy of Stoffle, Van Vlack, and Kays).

While archaeology studies of the canyon rim structures suggest they were all built with small holes in the walls to permit narrow shafts of light to shine on interior walls. Native representatives argue that these holes were for telling time (Figure 10). At some locations, even the buildings themselves cast shadows on each other at key astronomical moments. These shafts of light are generally understood as emanating from the sun, moon, planets, and stars. Simply put, the structures are timekeepers according to representatives from different tribes and pueblos:

*There has to be a connection. Did we want to sit on the mesa to be close to the heavens? To be close to the spirits? Somewhere, I heard that. I see why. You want to build something that is close to the heavens. That is your connection to the spirits.*

*There is a special group of people that stay on top of our mesa for the whole year, they get elected, and those are called our field chiefs. So they are the ones that upkeep the village. And I am pretty sure every village had somebody to stay around and do upkeep. Important people do that. And we still carry on that tradition today. They go to the outskirts of the village and do their pilgrimage and pray around. A lot of their pilgrimage is to the mountain as well. There had to be someone to stay [here] and I am pretty sure they would do the same thing.*

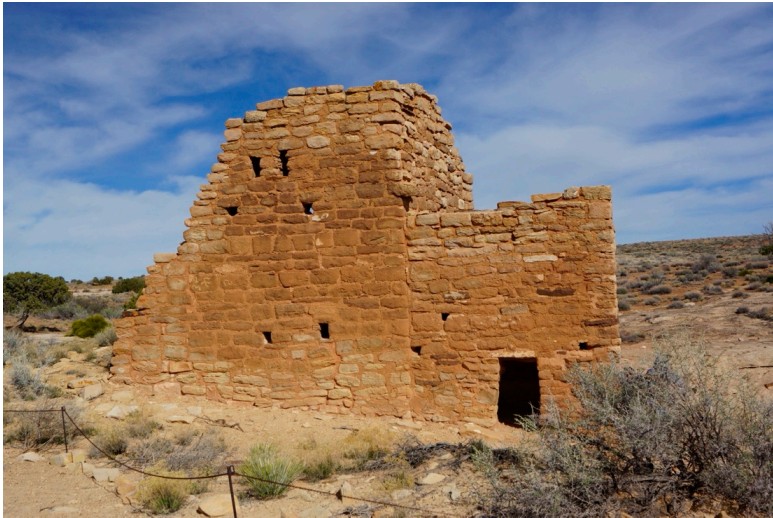

**Figure 10.** Time Keeping, Small Holes in Walls, Entrance Door in the Side (Photo Courtesy of Stoffle, Van Vlack, and Kays).

Many structures have specialized light capture angles, ceremonial kivas, multistory towers on top, and are built around massive stone boulders. Figure 11 is a square tower built on a free-standing stone at the head of a canyon at a spring. Note that there are small holes for time keeping it its walls. Figure 12 is a round tower that is built on and around large boulders at a canyon-head spring. A tunnel from its base goes to an underground kiva. Tribal and pueblo representatives interpret these as (1) having been placed on the large boulders in order to interact with the earth itself, (2) are multiple storied towers in order to gain heights to send prayers to the surrounding mountains, and (3) are near the spring to send and receive prayers from the water landscape that feeds the spring from the mountains. The canyon head and nearby areas provide special acoustics that produce a strong and unified sound from multiple kiva ceremonial prayer groups when they sing and drum at the same time.

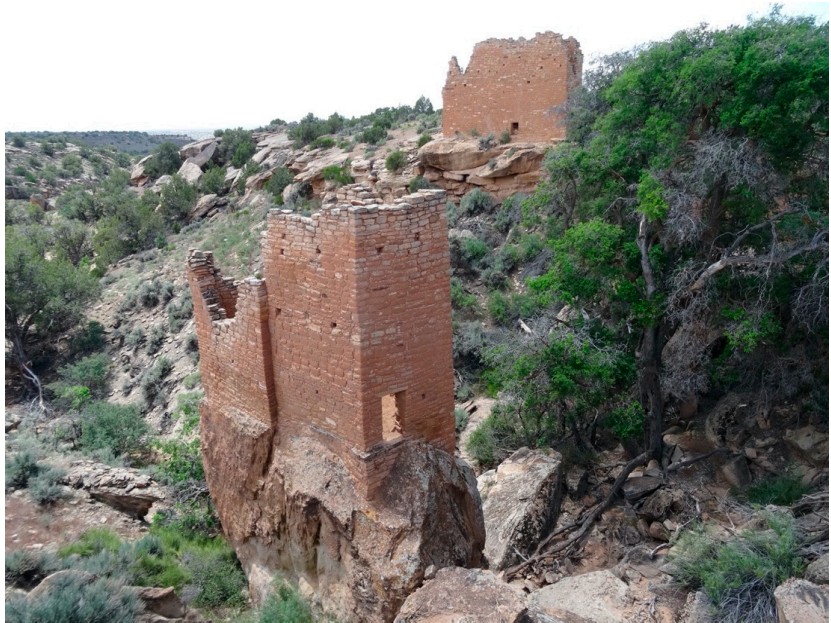

**Figure 11.** Hovenweep Tower on Boulder Near Spring (Photo Courtesy of Stoffle, Van Vlack, and Kays).

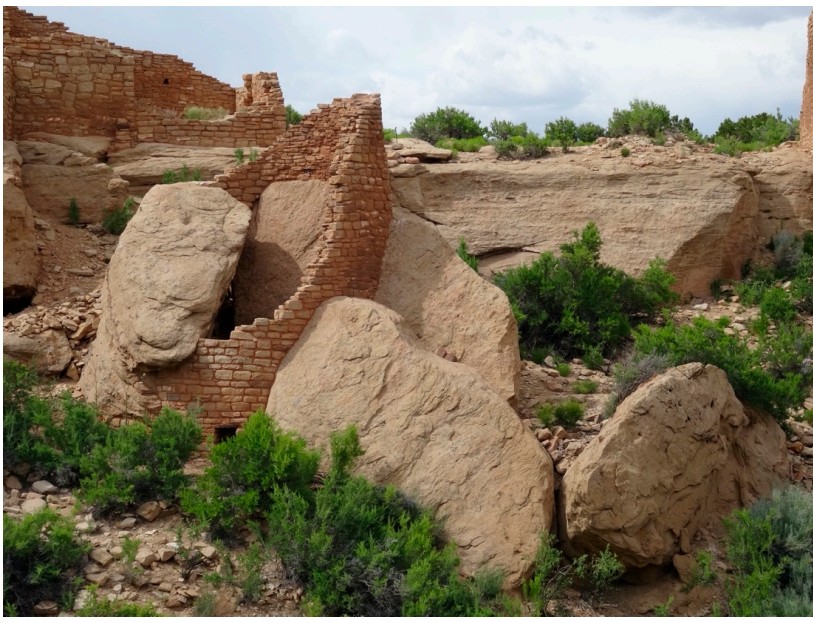

**Figure 12.** Hovenweep Tower Built Around Boulders, Near Spring, Kiva Below (Photo Courtesy of Stoffle, Van Vlack, and Kays).

*Recommendations*

The NPS has placed most canyon bottom kivas, towers, and springs off limits for tourists' visits, although they can be viewed from the canyon rims. This was a positive decision from the cultural perspectives of the representatives who maintain that none of these should be visited by tourists. However, two areas including springs with tower-kiva complexes and associated rim buildings are open to tourists. Neither has physical barriers nor signage. Representatives would like for these places to be closed like others in the park. Representatives also would like to return to the springs to clean them and say prayers, as is discussed above for Cave Spring.

## 8. Case 3: Arches Day and Night

The third heritage area is Arches, which has more than 2000 sandstone arches that are visually and ceremonially connected with distant places like Sky Islands massifs. Representatives identified these arches as portals to other dimensions [53]. Note in the foreground of Figure 13 that the NPS ended the developed packed earth trail at a considerable distance away from the arch itself in order to reduce tourist visitation inside the arch. This received positive comment from tribal representatives.

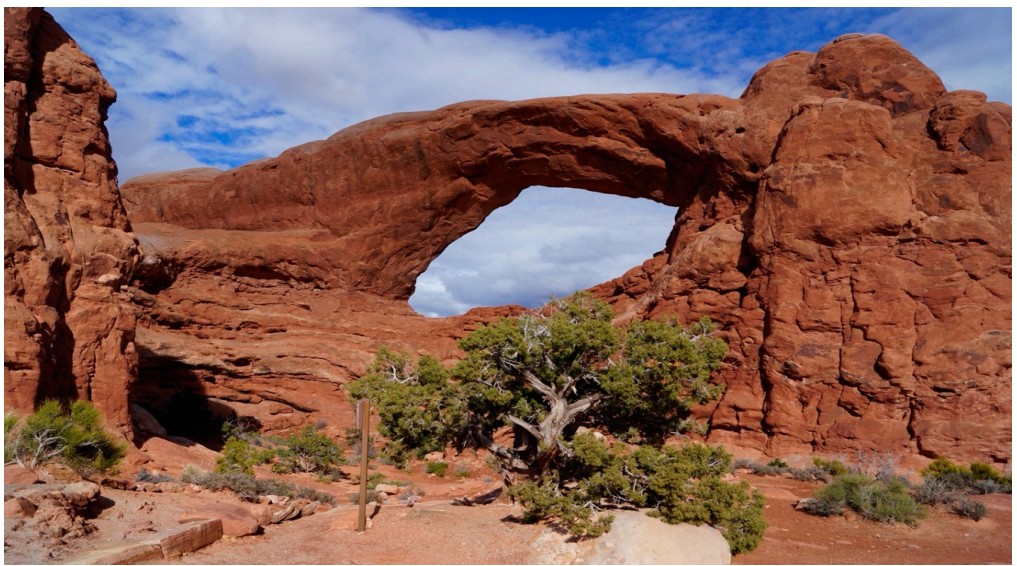

**Figure 13.** Arch as Portal (Photo Courtesy of Stoffle, Van Vlack, and Kays).

The arches serve(d) as portals or frames through which both travelers pass to more distant places and others receive information and energy from the sun, moon, and stars. These arches contribute to the meaning of, and derive meaning from, these visual and portal heritage landscapes. This was explained by one representative:

> *We have this place that is called the Lookout in the Grand Canyon, it is a four-wall site/structure that has four windows, north, west, south, and east directions, and it has the same use to areas like this. They use that first structure to look into the past, look into the future, looking to the directions where they can go. That structure was built by our ancestors, and when they came to these areas they had these [points at arch], it has the same power, the same meaning, they use these as windows into the future, to look into the direction they had to go. All the arches here, they all have the same power, the same significance, even if it's a very small arch, that is has that ability for people to look in. We came up, while we were talking, how our ancestors did it, it's tu-na-pi-quai, and it means looking through. If you look into the arch it's like looking into a glass, so it has that power, that ability, to give our people the sense of where to go.*

The park portion of these landscapes also contains hundreds of dramatic hoodoos which are perceived by the Native American representatives to be alive and capable of interacting with prayers

and healing ceremonies (see Figure 2 above). Hoodoos have a role in ceremony today, as explained by a representative:

> *We were discussing some of the hoodoos here and that we still have that same practice in Zuni. Our religious leaders actually put stone pillars in places where we need to leave offerings in the village. So looking at these pillars here, this might have been the place where they picked up that idea of using pillars to make the shrines. Just looking at all of these hoodoos, they're very powerful. So just being here gives a person, well for me, gives me a sense of connecting and looking into what our ancestors were doing here.*

Together with the thousands of arches, the hoodoos help define the Creation purpose of this area for world balancing ceremonies conducted by multiple religious specialists from distant communities. The contemporary heritage importance was discussed by another representative:

> *It's not a pretty rock it hosts something very significant for us. Because I mention that, if you look into the past and walk into areas like this, and realizing our great great great grandfathers and grandmothers also walked in areas within this place here, and so it holds a really strong bond with us because of that connection back then. Even though the sand shifted, the footprints so you can't identify them, but for us they're still here. We consider our ancestors as being a part of this, they never left. The people that passed on that are making their final journey into the afterworld were left here. And so with that understanding that we still have a very strong bond to places like this because our ancestors are still here, their remains are still here, so we never want to break the bonds with our ancestors because the more we travel, the more we get that connection everywhere back together again.*

Arches were involved in ceremonies conducted both day and night, the latter time being especially powerful for travel through the portal and for talking with the arch itself (Figure 14). Representatives said that many of the ceremonies associated with the arches would be conducted at night. This time is best for viewing celestial lights. Day and night transitions would be especially important as the sun sets and the moon rises, especially at key temporal events such as the winter and summer solaces and the vernal equinoxes. Other deep dark night activities would occur without the moon and be governed by the appearance of star formations and various planets. Thus, the cultural heritage landscapes of Arches National Park have both day and night dimensions. Such a study would contribute to growing literature on night tourism [54].

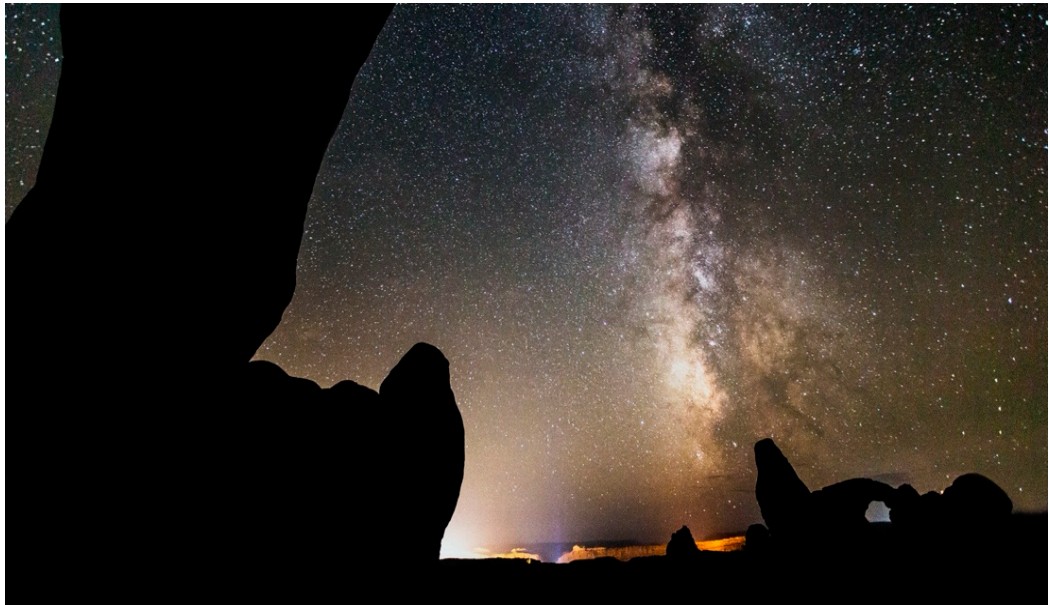

**Figure 14.** Night Sky at Arches National Park (Photo Courtesy of the National Park Service).

*Recommendations*

Representatives agree that arches should be approached by visitors with a knowledge of what they are and how to respectfully talk with them. Loud talking and throwing rocks should not occur under the arch. Generally, the space under the arch is too powerful for persons, especially children, who have not been culturally prepared, and so access should be restricted and certainly the tourists should be made aware of this condition.

Arches National Park, Canyonlands National Park, and Hovenweep National Monument have been declared as Certified International Dark Sky Parks (International Dark Sky Association IDA) so there are new kinds of tourists and activities occurring around, under, and through the arches. Night is a special time for native ceremonies. Dark Sky Park tourist activities should be formally studied in consultation with tribal and pueblo governments so that culturally sensitive protocols can be developed to assure sustainable tourism occurs.

## 9. Analysis

Native American representatives expressed the common hope that their traditional heritage places would be returned to the control of tribal and pueblo governments and subsequently qualified religious and spiritual leaders [55]. Unlike some countries, such as Australia and New Zealand, the U.S. has been slow to return control of federal lands to native peoples. There are, however, widespread instances of co-stewardship with the U.S. Department of Defense [56], the Nevada National Security Site, or Nevada Test Site, Department of Energy [57], and U.S. Fish and Wildlife Service [58]. The NPS co-manages some National Parks and Monuments with Native tribes, such as Pipe Spring National Monument. Recently the NPS established a nationwide initiative for sharing plants on parkland with culturally associated tribes under the new Plant Gathering Rule titled, "Gathering of Certain Plants or Plant Parts by Federally Recognized Indian Tribes for Traditional Purposes" [59].

Today, Native Americans control only a fraction of their aboriginal lands as reservation lands. Southern Paiutes, for example, have lost 98% of their aboriginal lands to encroachment [24]. Native Americans and other indigenous peoples stipulate that their aboriginal connections to lands were established at Creation, and this has since bound peoples and their ancestral lands in reciprocal obligations of nurturance and care [60]. Despite being removed, Native Americans continue to desire to strengthen their connections with aboriginal lands and to participate in their preservation.

The three EOAs in this analysis provided the NPS with better ideas of native voices regarding the meanings of native heritage resources being held, managed, and interpreted by the park. While the studies provided new ideas for incorporating Native American recommendations into how the parks present themselves to their millions of annual visitors, a separate set of intellectual and practical challenges are still to be addressed. Not the least of these challenges are *epistemological divides* created by the confrontation of different worldviews, such as whether or not stone arches are inert products of erosion or portals between spatial and temporal dimensions prepared at Creation for Indian spiritual leaders? Whether or not spring are portals to other dimensions as well as being connected with the snow and rain on top of neighboring Sky Island massifs? Whether or not prayer towers and kivas potentially still can serve to collectively produce and transport prayers for peace and balance in the world? Whether the functional integration of native heritage landscapes can be influenced by tourists' activities at places in the landscape? Resolving these differences in epistemology will be necessary in order to establish a foundation for new intellectual relationships between parks, Native American peoples, and tourists.

This analysis has focused on a small subset of Native American sacred places. Deloria and Stoffle [56] summarized the types of sacred places that are known to occur on federal lands. These types include:

1.   Creation Story Locations and Boundaries
2.   Sacred Portals Recounting Star Migrations

3. Universal Center Locations
4. Historical Migration Destiny Locations
5. Places of Prehistoric Revelations
6. Traditional Vision Quest Sites
7. Plant–Animal Relationship Locations
8. Mourning and Condolence Sites
9. Historical Past Occupancy Sites
10. Spirit Sites
11. Recent Historical Event Locations
12. Plant, Animal and Mineral Gathering Sites
13. Sanctified Ground

Only Native Americans can identify their heritage places and landscapes that occur in national parks and other nearby locations. The list of sacred heritage places commonly found on federal lands potentially broadens expectations regarding the kinds of issues that can emerge during consultation. These heritage sites are expected to be layered by time because Indian people have lived throughout North America for more than 17,000 years. Any given place that tourists visit can have multiple kinds of heritage effects for Native Americans.

## 10. Conclusions

Every day, U.S. national park employees strive to meet Congressional expectations that they will preserve and provide access to the natural and anthropogenic resources they manage. It is generally understood that most resources are more sustainable without tourists and yet tourists provide the funding and reason for the national parks. Therein lies the quandary; how to achieve both goals? Clearly these findings indicate that Native Americans and the National Park Service agree, based on their past experiences, that tourist education which includes a clear native perspective is the most viable alternative for preserving native heritage places and landscapes.

The problem is exacerbated because most Native Americans believe that their heritage resources exist in multiple dimensions of reality and are both physical and spiritual. Most Native Americans share an epistemology that all the world is alive and sentient and can be talked with regarding management issues. Sustainable use of these resources by tourists requires some consensus regarding these heritage issues.

This final discussion is structured around three questions. First, how common is it for tourists to disrupt or even destroy Native American sacred places? Second, will tourists follow NPS guidelines at Native American heritage sites in national parks. An extension of this question is whether or not it is reasonable to assume that tourists held to one standard for interacting with heritage sites in national parks will apply those standards to their behavior elsewhere? Third, is sustainable tourism development the best planning and management model for protecting native heritage sites?

### 10.1. Tourist Impacts to Heritage Sites

Tourist do impact places they visit either by just using the area, e.g., trampling, or by doing deliberate harm, e.g., applying graffiti [61]. Pandemic studies are now documenting an impact called *anthropause*, which is having a measurable and widespread positive benefit for nature because fewer people are on the land, including parks [62]. Tourists, however, also spend money, so the economic impacts of national parks on nearby communities is a strong motivator for their being established [63].

The NPS is constantly struggling to police and restore tourist impacts, while providing as much access as possible [64]. Chaco Culture National Historical Park, for example, closed many areas to tourists as a consequence of tourist impacts. The largest kiva in north America, called Casa Rinconada, and the famous Solar Calendar on Fajada Butte were closed due to visitor impacts. Our Native American study [65] documented tribal and pueblo recommendations that not every visitor should be

restricted, but instead rules should specify which tourists are most problematic and encourage others to continue visitation [66].

The initial large-scale site-intercept study regarding the interface between heritage tourism and Native Americans was conducted in four national parks located north of the Colorado River [11,66]. A total of 1796 completed interviews were conducted with touring persons in Zion National Park, Pipe Springs National Monument, Lees Ferry in Glen Canyon National Recreation Area, and the north rim of the Grand Canyon National Park. The findings provide ideas about kinds of heritage tourists, their potential impacts, and their expenditures. They were primarily nature/environmental tourists, with about 2% primarily attracted to the parks by native culture. When asked what they viewed as most interesting about visiting a native reservation, it was their culture today. People who neither wanted to damage the reservation environment nor intrude on normal native life tended to spend more per day than others on native curios. Tourists holding positive attitudes towards native people, however, were less likely to return if their heritage education goals were not achieved.

Arches has 1.3 million tourists a year. The NPS reduced adverse tourist impacts through a system of well-developed trails and clear signage about staying on the established paths to reduce erosion and protect the crust soils. Well trained and armed rangers patrol the different kinds of arches designations all of whom are reached by approved trails. Leaving NPS trails to walk across the soft earth, however, soon results in adverse impacts including a deepening of the new trails. Tourist visits under arches have potentially spiritual effects; thus reducing the willingness of the arch to talk with native peoples.

In Arches, the recommendations for interpretating stone arches as portals have not been adopted, so it is too early to see how restricting tourists might work. In Canyonlands, the Cave Spring site has been cleaned and new tourist barriers have been erected. Hovenweep control over tourists already occurs due to well-managed paved, packed, and clearly marked trails. As a small park, Hovenweep is vulnerable to losing staff positions, especially trained ranger police and guides. So tourist control at distant monument units is challenging.

*10.2. Compliance with NPS Guidelines*

There is an expanding literature that has tested the outcomes of interpretation programs on tourist behaviors [67]. Littlefair and Buckley [68] developed an experimental methodology in an Australian World Heritage Area for testing the impacts of interpretation. They found that interpretation by skilled guides can reduce visitor impacts in protected areas, especially if role modeling is combined with verbal appeals.

Native Americans who participated in the three EOA studies are aware from past consultations that when tourists are educated by the park, they do less harm to native heritage resources. NPS funds and trained staff are required to interpret and educate millions of tourists, who mostly come to iconic parks. Once educated by NPS staff and having experienced culturally appropriate behaviors in iconic parks, it is assumed that these tourists will transfer appropriate behaviors elsewhere when touring less protected heritage landscapes and smaller parks.

*10.3. Sustainable Tourism Development*

Richard Sharpley [69,70] published two research-based papers in the *Journal of Sustainable Tourism*; both of which concluded based on thousands of publications that it is not possible to simultaneously pursue the goals of sustainable tourism and sustainable development. He argues that environmentally sound tourism is possible, but with a sustainable de-growth approach. He concludes that there is little evidence of progress towards the achievement of sustainable tourism development as a combined effort. He suggests un-coupling sustainable tourism and development because it is an unachievable model, and to pursue instead sustainable tourism with more modest economic goals.

Certainly, our study of the sustainable use of Native American heritage places cannot resolve the thousands of data driven discussions of sustainable tourism development. This analysis, however,

does lend a voice to the issue of considering native heritage places and landscapes as separate variables for assessing sustainable tourism development.

Removing or reducing economic considerations from national park planning and management is problematic for native heritage site protection. Large iconic parks produce funds that help support smaller parks, many of which have vulnerable native heritage places. Moreover, because of iconic parks like Arches, regional NPS offices like the Southeast Group can be supported. Multi-park management offices use personnel supported by iconic parks to help manage smaller parks such as Hovenweep NM. Few small parks and monuments could successfully protect native heritage places and landscapes without the economic benefits of tourists who are attracted to iconic parks. Both the participating Native Americans and the staff of these national parks work within a model which plans for expanding tourist visitation; however, they also share the belief that tourist education will reduce damage to these native heritage places and landscapes.

**Author Contributions:** R.S. served as the principal investigator for all three EOA studies. K.V.V. served as a lead researcher on the Arches and Canyonlands EOA. C.K. served as a project researcher on the Hovenweep EOA; conducting interviews, analyzed data, and helping write the final report. O.S. is a member of the Pueblo of Zuni and is the Director of the Zuni Cultural Resources Advisory Team. He participated in all three EOA studies as an official representative of the Pueblo of Zuni. All authors have read and agreed to the published version of the manuscript.

**Funding:** Funds for three studies were provided to the Regents of the University of Arizona, through a grant with Cooperative Ecosystem Study Unit—National Park Service. The Southeast Utah Group, Moab, Utah of the NPS provided considerable support.

**Acknowledgments:** The authors acknowledge our nine tribal and pueblo partners who participated in these EOA studies: Acoma Pueblo, the Hopi Tribe, the Kaibab Band of Paiute Indians, the Navajo Nation, the Paiute Indian Tribe of Utah, the Pueblo of Zuni, Santa Clara Pueblo, Southern Ute Indian Tribe, and the Ute Indian Tribe. Their dedication and openness made the studies possible. NPS staff contributed to the success of these EOA studies. They included Karen Wurzburger, Chris Goetze, Laura Martin, Mark Miller, Terry Fisk, Kelly Adams, Thann Baker, and Noreen Fritz. We also extend our gratitude to the applied ethnographic research team at the School of Anthropology, University of Arizona. Helping with these EOAs were professor N. Pleshet; post-doctoral research associates K. Van Vlack and K. Brooks; graduate students E. Pickering, C. Sittler, H. Lim and M. Albertie; and undergraduate students M. Johnson, C. Kays, G. Penry, and C. Forer. Professor Diane Austin was Chair of the School of Anthropology during these studies and Jeannine McElveen was the Superintendent who facilitated the studies at Hovenweep, Kate Canon was the Superintendent at Canyonlands and Arches National Parks who helped facilitate these studies.

**Conflicts of Interest:** The authors declare no conflict of interest.

## Appendix A. The BARA Methodology

The Bureau of Applied Research in Anthropology (BARA) was founded by the state of Arizona and placed in the University of Arizona in 1952 as the Bureau of Ethnic Research (BER). The BER was charged with the responsibility to monitor the socio-economic welfare of Native American communities in Arizona. In 1982, the BER changed its name and vastly expanded its research and training mission. The BARA faculty has been comprised of sixteen state-funded and project funded academic professionals organized around six different programs. For each program, there exists a set of research activities consistent with the BARA mission, as well as corresponding academic courses and student participation that contribute in an integrated fashion to BARA's commitment to applied training.

The BARA ethnographic team involved with these EOA study directs a program called Native American Cultural Resource Revitalization. Consistent with BARA's founding mission, to monitor the welfare and well-being of Native American groups in Arizona, this program focuses on the national need to assure the preservation of Native American cultures and languages. A long history of misguided policy making and disregard for native cultures in this country has created marginalized and dependent peoples with severe economic disadvantages and little control over their own destiny. Legislation, such as the American Indian Religious Freedom Act of 1978 and the Native American Graves Protection and Repatriation Act of 1990, has attempted to redress the situation and establish

new policy paths that emphasize tribal empowerment and cultural respect. BARA has contributed to these new directions by developing standard procedures that assure the full participation of Native American tribes in the process of identifying and controlling their comprehensive cultural resource inventories. In this program, BARA researchers facilitate the interaction of tribes with government agencies and private organizations. Through the use of ethnography, BARA professionals have assisted communities in the reconstruction of their cultural histories, made Geographic Information Systems (GIS) technologies available to tribes wanting to identify and maintain their cultural landscapes, and worked to address language shift through the development of dictionaries and the promotion of language literacy on reservations.

This program has also contributed to the development of cultural resource theory within applied anthropology and has generated genuine, mutually respectful, and productive partnerships between the university and Native American tribes. One of BARA's most consistently supported research programs, the Native American Cultural Resource Revitalization, has received long term funding from tribes, the National Park Service, the Department of Energy, the Department of Defense, the Bureau of Reclamation, National Science Foundation, and other entities.

*Appendix A.1. Summary of BARA Interview Instruments*

The following is a brief discussion of four survey instruments used by the BARA research team in these three EOA studies. Brief paragraphs explain the purpose behind each survey instrument and the types of information they seek to ascertain.

Appendix A.1.1. Site Form

The Site Interview form is *place-specific* and is used to record site use history and types of ethnographic resources associated with site use including water, plants, animals, minerals, landforms, and archaeological remains. With this form, the ethnographer can elicit detailed information on material, behavioral, and spiritual connections among resource types, and between each resource and a place. It was used initially in Zion National Park and Pipe Spring National Monument Study [38,39]. The "Zion form" has since been successfully applied in numerous federally funded projects that involved tribes in the West and Midwest regions of the United States.

Appendix A.1.2. Ethnoarchaeology-Rock Art Form

The second type of survey instrument is called the Rock Art Form. It is used in the event that the Site Interview is too general, and more fine-grained analysis is feasible and useful for a study. This form is specifically used when petroglyphs or pictographs are the dominant resource at the site. The Rock Art form focuses on an individual's and ethnic group's use, meaning of rock art panels, and their understanding of how it is connected to the surrounding landscape [71,72].

Appendix A.1.3. Cultural Landscapes Form

The Cultural Landscape Form was designed with input from agencies who needed to have a way to manage much larger areas as integrated cultural phenomena and with Indian people whose culture is organized in terms of such big areas. At a national federal policy level such efforts correspond with the concept of Ecosystem Management [73]. The landscape form frames place and resource-specific information in a broader regional and more abstract cultural context. With this form, we investigate origin and migration traditions, ethnic group settlement and land use history, and specific use patterns of the natural topography. Data on trail systems, including travel across land and through water, and ceremonial trails associated with songs, drum circles, dreaming, pilgrimages, and individual quests, are also crucial to unraveling complex cultural connections between places and resources.

Appendix A.1.4. Cultural Landscapes—Pilgrimage Connections Form

The interconnectedness of places is very important for understanding how Indian people view the landscape. This key element presents an opportunity to see specifically how ceremonial sites are connected to each other. To explore this issue in detail, a form was developed to provide Indian people with an opportunity to see if places already visited and evaluated by them are connected. Once Indian people establish that the places are connected, they are then asked to draw the perceived pilgrimage trails, or Puha Paths, a vision quester would travel to the ceremonial destination site. This form was first used during the Black Mountain Ethnographic Study on Nellis Air Force Base [74].

*Appendix A.2.*

**Native American Ethnographic Resources**

**Site Form**

**University of Arizona Indian Note Form**
* You must record a response for every question asked in order for data to be correctly coded *
**Interview Number**: ________________

1. Date:________________
2. Respondent's Name: ________________________________
3. Tribe/Organization:          3a. Ethnic Group:
4. Gender:  Male   Female
5. Date of Birth: ___/___/___ 5a. Age ______
6. Place of Birth (Town, Reservation): ________________ 6a. U.S. State of Birth ________________
7. Study Area Site Number (ethnographer fill this in): ________________________
8. What is the name of this place in English?  8a.  What is the name of this place in your native language?
9. Please describe the geography of this area or elements which stand out.
10. Would Indian people have used this area? 1 = YES 2 = NO 8 = Don't Know 9 = No Response
10a. (IF YES) Why or for what purpose would Indian people have used this area? 1 = [permanent]LIVING 2 = HUNTING 3 = [seasonal]CAMPING 4 = CEREMONY/POWER 5 = GATHERING FOOD 6 = OTHER 8 = Don't Know 9 = No Response
10b. Comments on 10a:
11. Is this place part of a group of connected places (Is this place connected to others?)  1 = YES 2 = NO 8 = Don't Know 9 = No Response
11a. (IF YES) What kinds of other places might this place be connected with and where are they? 1 = Comment given 8 = Don't Know 9 = No Response
11b. (IF COMMENT GIVEN) How is this place connected to the others you mentioned? 1 = Comment given 8 = Don't Know 9 = No Response
11bb.(IF ANSWERED 1 TO 11b) Comments given:

Which, if any, of the following features is an important part of why this place is significant to Indian people?

**For Each Feature Please Fill Out Appropriate Feature Page**

FEATURE TYPE A: WATER SOURCE (List specific feature from Table A1 on page 3).

**Table A1.** Place Features (Explain you will now begin asking questions about the physical features of the place).

| Feature Type | 1 = YES | 2 = NO | List and Describe Each Specific Feature |
|---|---|---|---|
| 12a. Source for Water | | | 12aa. |
| 12b. Source for Plants | | | 12bb. |
| 12c. Source for Animals | | | 12cc. |
| 12d. Evidence of Previous Indian Use e.g.,- rock rings, historic structures, rock art | | | 12dd. |
| 12e. Geological Features e.g.,- mountain, spring, cave, canyon, landmarks | | | 12ee. |

13. Would Indian people have used this (Name the feature)? 1 = YES 2 = NO 8 = Don't Know 9 = No Response

14. (IF YES) Why or for what purpose would Indian people have used this __Feature(s)__? 1 = FOOD/DRINK 2 = MEDICINE 3 = CEREMONY 4 = OTHER 8 = Don't Know 9 = No Response

14a. Comments:

15. How would you evaluate the condition of the ___Feature(s)__? 1 = EXCELLENT 2 = GOOD 3 = FAIR 4 = POOR 9 = No Response

16. Is there anything affecting the condition of the __Feature(s)__? 1 = YES 2 = NO 8 = Don't Know 9 = No Response

16a. (IF YES) What in your opinion, is affecting the condition of ____________? FEATURE TYPE B: PLANT SOURCE (List features from Table A1 on page 3)

17. Would Indian people have used the plants at this particular site? 1 = YES 2 = NO 8 = Don't Know 9 = No Response

18. (IF YES), Why or for what purpose would Indian people have used these plants? 1 = FOOD 2 = MEDICINE 3 = CEREMONY 4 = MAKING THINGS 8 = Don't Know 9 = No Response

18a. Comments (if given):

19. How would you evaluate the condition of these plants? 1 = EXCELLENT 2 = GOOD 3 = FAIR 4 = POOR 9 = No Response

20. Is there anything affecting the condition of these plants? 1 = YES 2 = NO 8 = Don't Know 9 = No Response

20a. IFYES) What in your opinion, is affecting the condition of the plants?

Feature Type C: Animal Source (List features from Table A1 on page 3)

21. Would Indian people have used the animals at this place? 1 = YES 2 = NO 8 = Don't Know 9 = No Response

22. Why or for what purpose would Indian people have used the animals in this site? 1 = FOOD 2 = MEDICINE 3 = CEREMONY 4 = CLOTHING 5 = TOOLS 6 = OTHER 8 = Don't Know 9 = No Response

22a. Comments:

23. How would you evaluate the condition of these animals/habitat? 1 = EXCELLENT 2 = GOOD 3 = FAIR 4 = POOR 9 = No Response

24. Is there anything affecting the condition of the animals/habitat? 1 = YES 2 = NO 8 = Don't Know 9 = No Response

24a. (IF YES) What in your opinion, is affecting the condition of the animals/habitat? FEATURE TYPE D: EVIDENCE OF PREVIOUS OCCUPATION OR USE (Specifically)

25. Would Indian people have used this site and/or artifacts? 1 = YES 2 = NO 8 = Don't Know 9 = No Response

26. Why or for what purpose would Indian people have used this site and/or artifacts? 1 = LIVING 2 = HUNTING 3 = GATHERING 4 = CAMPING 5 = CEREMONY/POWER 6 = OTHER 8 = Don't Know 9 = No Response

26a. Comments:

27. How would you evaluate the condition of this site?  1 = EXCELLENT 2 = GOOD 3 = FAIR 4 = POOR 9 = No Response

28. Is there anything affecting the condition of this site?  1 = YES 2 = NO 8 = Don't Know 9 = No Response

28a. (IF YES) What in your opinion, is affecting the condition of this site? FEATURE TYPE E: GEOLOGIC FEATURES (specifically)

29. Would Indian people have visited or used this __(Feature)__ ? 1 = YES 2 = NO 8 = Don't Know 9 = No Response

30. Why or for what purpose would Indian people have used this __(Feature)__ ? 1 = SEEK KNOWLEDGE/POWER 2 = COMMUNICATE WITH OTHER INDIANS 3 = CEREMONY 4 = COMMUNICATE WITH SPIRITUAL BEINGS 5 = TEACHING OTHER INDIANS 6 = TERRITORIAL MARKER 7 = OTHER 8 = Don't Know 9 = No Response

30a. Comments:

31. How would you evaluate the condition of the __(Feature)__? 1 = EXCELLENT 2 = GOOD 3 = FAIR 4 = POOR 9 = No Response

32. Is there anything affecting the condition of the __(Feature)__? 1 = YES 2 = NO 8 = Don't Know 9 = No Response

32a. (IF YES) What in your opinion, is affecting the condition of __(Feature)__? MANAGEMENT AND ACCESS RECOMMENDATIONS

33. How would you evaluate the condition of this place?  1 = EXCELLENT 2 = GOOD 3 = FAIR 4 = POOR 9 = No Response

34. Is there anything affecting the condition of this place?  1 = YES 2 = NO 8 = Don't Know 9 = No Response

34a. (IF YES) What in your opinion is affecting the condition of this place? Above you identified specific features at this site. What would be your recommendation for protecting each specific feature?

35. Water Source:

36. Plant Source:

37. Animal Source:

38. Traditional Use Feature:

39. Geological Feature:

40. What would be your recommendation for protecting this place?

41. Do you think Indian people would want to have access to this place?  1 = YES 2 = NO 8 = Don't Know 9 = No Response

41a. (IF YES) Why would Indian people want to come to this place? Are there any special conditions that must be met for Indian people to use this place? 1 = YES 2 = NO 8 = Don't Know 9 = No Response

42a. (IF YES) What special conditions are needed for Indian people who want to come to this place?

**Comments:**

**Native American Ethnographic Resources**
**Ethnoarchaeology-Rock Art Questions**

University of Arizona Interview Form
Date: _________　　　　　　　Interview #: _________
Interviewer: ____________

2. Respondent's Name: ____________

3a.　Tribe: _______　　2b. Ethnic Group: ______

4.　　Gender: (Circle)　　*1 = M*　　*2 = F*

5a.　English Name of site _______ 4b. Site No.: NV _______

5e.　Quad Name _______ 4f. Compass Orientation _______ 4g. Elevation ______

6a.　Study Area Site #_________ 6b. Ecozone Location: 6c. Topography: 6d. Main Water Source:

i.　　canyon wall　　　　　　i. delta　　i. River edge

ii.　　UDSZ-desert　　i. side canyon　ii. River flood

iii.　　OHWS-old riparian　iii. wash or drain　iii. Side stream

iv.　　REPS-new riparian　iv. mesa top　　iv. Spring

v.　　side canyon riparian　v. canyon wall　v. Rainfall

vi.　　dry mesa top　　vi. saddle　　vi. rock tank

vii.　　stream bed　　vii. talus　　vii. wash

viii.　high desert flats　viii. cave

ix.　　upper Mohave desert

x.　　lower Mohave desert

xi.　　stream bank

xii.　　woodland

7.　　Did you know that **this site** was here?
　　　*1 = Yes*　　*2 = No*　　*8 = DK*　　*9 = NR*

**Ethnic Group Use History**

500.　In your opinion, was/were (**this/these panel(s)**) made by your people? *1 = Yes*　　*2 = No*
　　　*8 = DK*　　*9 = NR*

501.　Did your [respondent's ethnic group] traditionally visit or use (**this/these panel(s) or panel(s) like this/these [where?]**)? *1 = Yes*　　*2 = No*　　*8 = DK*　　*9 = NR*

502.　IF YES TO #501) What were (**this/these panel(s) or panel(s) like this/these**) visited or used for? *1 = Ceremony (SPECIFY)*　　*2 = To Seek Knowledge/Power*　　*3 = Communicate w/Other Indian People*　　*4 = Communicate with Spiritual Beings*　　*5 = Teaching Other (ethnic group) People 6 = Territorial Marker 7 = Decoration*　　*8 = Other (SPECIFY)*　　*9 = Map*　　*10 = paying respects 11 = N/A*

502b. What kind of name would you give this panel?

503.　Who visited or used (**this/these panel(s) or panel(s) like this/these**) most often? *1 = Men 2 = Women　3 = Both　7 = NA　8 = DK　9 = NR*

504.　Do your people currently visit or use (**this/these panel(s) or panel(s) like this/these [where?]**)? *1 = Yes　2 = No　7 = NA　8 = DK　9 = NR*

505.　(If yes to #504) What are (**this/these panel(s) or panel(s) like this/these**) visited or used for? CIRCLE BELOW *1 = Ceremony (SPECIFY)*　　*2 = To Seek Knowledge/Power*　　*3 = Communicate w/Other Indian People*　　*4 = Communicate with Spiritual Beings*　　*5 = Teaching Other (ethnic group) People*　　*6 = Territorial Marker*　　*7 = Decoration*　　*8 = Other (SPECIFY) 9 =·Map*　　*10 = paying respects*　　*11 = N/A*

506.　Who visits or uses (**this/these panel(s)/panel(s) like this/these**) most often? *1 = Men　2 = Women 3 = Both　7 = NA　8 = DK　9 = NR*

　　　CONNECTIONS

517.　Are there (ethnic group stories and legends associated with (**this/these panel(s) or panel(s) like this/these**) *1 = Yes　2 = No*　　*8 = DK　9 = NR*

517a. If YES, What is the name of that story?

517b. Can it be told to outsiders?　*1 = Yes*　　*2 = No*　　*8 = DK*　　*9 = NR*

517c. IF YES, will you tell us about that story? (make sure to record on tape)

518. I would like to ask you about the connections between (**this/these panel(s)** or **panel(s) like this/these**) and other resources.

519a. Are the pecking/paintings in this panel and connected in anyway with panels elsewhere? *1 = Yes* *2 = No*　　*8 = DK*　*9 = NR*

520a. Where and how are they connected?

519b. Are archaeology sites connected with these panels?　*1 = Yes*　　*2 = No*　　*8 = DK*　　*9 = NR*

520b. How are they connected?

519c. Are plants connected with these panels?　　*1 = Yes*　　*2 = No*　　*8 = DK*　　*9 = NR*

520c. How are they connected?

519d. Are animals connected with these panels?　　*1 = Yes*　　*2 = No*　　*8 = DK*　　*9 = NR*

520d. How are they connected?

519e. Are minerals connected with these panels?　　*1 = Yes*　　*2 = No*　　*8 = DK*　　*9 = NR*

520e. How are they connected?

519f. Is water connected with these panels?　　*1 = Yes*　　*2 = No*　　*8 = DK*　　*9 = NR*

520f. How is it connected?

519g. Is the surrounding land (geography, topog.) connected with these panels?　*1 = Yes*　　*2 = No* *8 = DK*　　*9 = NR*

520g. How is it connected?

HISTORY OF ETHNIC USE

521. Did Indian people who are not (your ethnic group) use (**this/these panel(s)** or **panel(s) like this/these**)? *1 = Yes*　*2 = No*　　*8 = DK*　*9 = NR*

522a. (IF YES TO #521) Who were those Indian people?

522b. Did those people use (**this/these panel(s)** or **panel(s) like this/these**) [*before, after, same time as*] respondent's ethnic group?　*1 = Before*　　　*2 = After*　　　*3 = Same time as* *4 = All of above*　　*7 = NA*　　　*8 = DK*　　　*9 = NR*

SEASONALITY, USE AND MEANING

523. Is there a special time of the year during which (**this/these panel(s)** or **panel(s) like this/these**) were/are used? *1 = Yes*　*2 = No*　　*8 = DK*　*9 = NR*

524. (IF YES TO #523) What special time of the year?

525. Is there a special time of day/night during which (**this/these panel(s)** or **panel(s) like this/these**) were/are used?　*1 = Yes*　　*2 = No*　　*8 = DK*　　*9 = NR*

526. (IF YES TO #525) What special time of day/night?

527. Within the site, are there any specific peckings or paintings that you would like to talk about? (IF NO, GO TO 530) *1 = Yes*　*2 = No*　　*8 = DK*　*9 = NR*

527a. Location of pecking/painting (boulder #) ____________

527b. Photo # (roll, shot)

527c. Indian/English name for pecking/painting ____________

527d. What is special about this pecking/painting?

528. Within the site, are there any other specific peckings or paintings that you would like to talk about? (IF NO, GO TO 530) *1 = Yes*　*2 = No*　　*8 = DK*　*9 = NR*

528a. Location of pecking/painting (boulder #) ____________

528b. Photo # (roll, shot)

528c. Indian/English name for pecking/painting __________

528d. What is special about this pecking/painting?

529. Within the site, are there any other specific peckings or paintings that you would like to talk about? (IF NO, GO TO 530) *1 = Yes    2 = No       8 = DK    9 = NR*

529a. Location of pecking/painting (boulder #) __________

529b. Photo # (roll, shot)

529c. Indian/English name for pecking/painting __________ ; __________

529d. What is special about this pecking/painting?

**(Back to general Discussion of site)**

530. Based on the rock art that you see at the site and on the ground, what Indian activities or events occurred at this site? [specify site]

531. Does the site have a personal meaning for you? *1 = Yes       2 = No       8 = DK       9 = NR*

532. (IF YES TO #531) What does the site mean to you?

533. How would you evaluate the overall importance of the site to you? *1 = Low    2 = Medium    3 = High    9 = NR*

534. Does the style of the peckings/paintings influence the cultural significance of this rock art? *1 = Yes    2 = No    8 = DK    9 = NR*

535. (IF YES TO #534) How?

*IMPACT ASSESSMENTS*

536. In your opinion, what is the current condition of this site? *1 = Excellent    2 = Good    3 = Fair    4 = Poor    8 = DK    9 = NR*

537. Do you feel there are human activities affecting the condition of the panels? *1 = Yes    2 = No    8 = DK    9 = NR*

537a. (if Yes to # 537) What human activities are affecting the condition of the panel(s)?

538. What would be your recommendation (if any) for protecting the panel(s) from human activities?

539. Do you feel there are natural elements (wind, rain, erosion) affecting the condition of the panel(s)? *1 = Yes    2 = No       8 = DK    9 = NR*

540. (IF YES TO #539) What natural elements are affecting the condition of the panel(s)? [specify general weather, other]

541. What would be your recommendation (if any) for protecting the panel(s) from natural elements?

542. Can you tell me anything else about the importance of (**this/these panel(s)** or **panel(s) like this/these**) to (respondent's ethnic group) that we haven't talked about?

**Native American Ethnographic Resources**

**The University of Arizona in Tucson**

**Landscape Questions**

**—Use along with map so people can point at places they talk about**

      ** NOTE: You must record a response for every question asked in order for data to be correctly coded- blank spaces are not responses ***

      Interview Number: _____________ Tape Number _____________ Date:_____________

      Respondent's Name: _______________________________

      Tribe/Organization: _____________________ Ethnic Group: _________________________

      Gender:   Male   Female

      Date of Birth: ___/___/___ Age _____

      Place of Birth (Town, Reservation): _________________ U.S. State of Birth _____________

      Study Area/place of interview (ethnographer fill this in): _____________________________

(1)     Were there Indian villages in relation to this area?

a.      1 = Yes, 2 = No, 8 = Don't Know, 9 = No Response.

(2)     If yes, were the area villages connected with villages elsewhere in the Southern Nevada/California region?

a.      1 = Yes, 2 = No, 8 = Don't Know, 9 = No Response.

(3)     If yes, how were these connected?
(4)     Do you know what the Indian people did when they were here in the area?

a.      1 = Yes, 2 = No, 8 = Don't Know, 9 = No Response.

(5)     If yes, what kinds of activities -
(6)     farming
(7)     gathering plants
(8)     gambling
(9)     ceremonies
(10)    political meetings
(11)    others (specify)
(12)    Do you know of Indian trails that were connected with this area?

a.      1 = Yes, 2 = No, 8 = Don't Know, 9 = No Response.

(13)    If yes, can you tell me something about those trails - like

a.      *   where did they go,   *   why did your people travel the trails, and

(14)    were these trails somehow special to your people? How?
(15)    Do you know of any songs associated with this area?

a.      1 = Yes, 2 = No, 8 = Don't Know, 9 = No Response.

(16)    If yes, can you tell me something about the songs—were they
(17)    traveling songs
(18)    ceremony songs, or
(19)    other-purpose songs
(20)    Do you know of any ceremonies that were conducted at or near this area ?

a.      1 = Yes, 2 = No, 8 = Don't Know, 9 = No Response.

(21)    If yes, can you tell me something about these ceremonies?

a.      * Ceremony #1—place ________________, when ______________, why ___________
b.      * Ceremony #2—place ________________, when _____________, why ___________
c.      * Ceremony #3—place ________________, when _____________, why _____________

(22)    Is this area at or near the place where your people were created?

a.      1 = Yes, 2 = No, 8 = Don't Know, 9 = No Response.

(23)    If yes, where is the Creation place?
(24)    Do you know if there are other places in this region that are also connected with the Creation of your people?

a.　　1 = Yes, 2 = No, 8 = Don't Know, 9 = No Response.

(25)　If yes, what and where are those places?

(26)　Do you recall or have your heard about events in history that occurred at or near this area?

a.　　1 = Yes, 2 = No, 8 = Don't Know, 9 = No Response.

(27)　Will you tell me something about those events?

(28)　Event #1—date ___________, place ___________, what happened?

(29)　Event #2—date ___________, place ___________, what happened?

(30)　Event #3—date ___________, place ___________, what happened?

(31)　Is there a connection between this area and nearby mountains?

a.　　1 = Yes, 2 = No, 8 = Don't Know, 9 = No Response.

(32)　If yes, what mountains and how are they connected to this area?

(33)　Mt. #1: name in English ______________, name in native language______________, how connected?

(34)　Mt. #2: name in English ______________, name in native language ______________, how connected?

(35)　Mt. #3: name in English ______________, name in native language ______________, how connected?

(36)　Is there a connection between this area and any section of the Colorado River?

a.　　1 = Yes, 2 = No, 8 = Don't Know, 9 = No Response.

(37)　If yes, what section of the river and how is it connected to this area?

(38)　River Section #1: name in English ______________, name in native language __________, how connected?

(39)　River Section #2: name in English ______________, name in native language __________, how connected?

(40)　River Section #3: name in English ______________, name in native language __________, how connected?

(41)　Is there a connection between this area and any of the creeks, springs, or washes in the Spring Mountains area (i.e., Deer Creek, Willow Creek, Lovell Wash, Wheeler Wash, Crystal Springs, etc.)?

a.　　1 = Yes, 2 = No, 8 = Don't Know, 9 = No Response.

(42)　If yes, which creeks, springs, or washes and how are they connected to this area?

(43)　Water source #1: name in English ______________, name in native language __________, how connected?

(44)　Water source #2: name in English ______________, name in native language __________, how connected?

(45)　Water source #3: name in English ______________, name in native language __________, how connected?

(46)　Is this area connected to any places or events in the region that we have not already talked about?

a.　　1 = Yes, 2 = No, 8 = Don't Know, 9 = No Response.

(47)　If yes, what other connections would you like to talk about?

(48)　Connection #1—place __________, event ____________, connection ___________

(49) Connection #1—place __________, event ____________, connection ___________

(50) Connection #1—place __________, event ____________, connection ___________

(51) Is this area connected to any places or events in your traditional territory that we have not already talked about?

a.　　1 = Yes, 2 = No, 8 = Don't Know, 9 = No Response.

(52) If yes, what other connections would you like to talk about?

(53) Connection #1—place __________, event ____________, connection ___________

(54) Connection #1—place __________, event ____________, connection ___________

(55) Connection #1—place __________, event ____________, connection ___________

**Native American Ethnographic Resources**

**Cultural Landscapes—Pilgrimage Connections Questions**

**University of Arizona**

**Interview Number: ______　　Tape: ______　　Ethnographer: _________**

1.　Date: __________
2.　Respondent's Name: ____________
3.　Tribe/Organization: ____________　　3a. Ethnic Group: ________
4.　Gender:　Male　Female
5.　Date of Birth: ____/____/____　　　　5a. Age _____
6.　Place of Birth (Town, Reservation): ________　　6a. U.S. State of Birth ______
7.　Study Location (ethnographer fill this in): ________
8.　What is the name of this area in English?　　　8a.　What is the name of this area in your native language?

　　The following questions provide an opportunity to see if places already visited and evaluated by Indian people are connected. When the person perceives that two or more places are connected, we would like to explore the nature of these connections. It is also important to know if the places are connected by a spiritual or physical trail, and if so, whether places are visited in sequence.

**Local Cultural Landscapes Questions**

(1), Do you think the places that you have visited in this area are connected?
1= Yes ______, 2= No ______, 8= Don't Know ______, 9= No Response ______
If so, which ones are connected.
Place __________ to place _____________; connected physically_____ or spiritually ______ or both.
Place __________ to place _____________; connected physically_____ or spiritually ______ or both.
Place __________ to place _____________; connected physically_____ or spiritually ______ or both.
Place __________ to place _____________; connected physically_____ or spiritually ______ or both.
Place __________ to place _____________; connected physically_____ or spiritually ______ or both.
Place __________ to place _____________; connected physically_____ or spiritually ______ or both.
Place __________ to place _____________; connected physically_____ or spiritually ______ or both.
(2), If these places are connected, were they used in sequence?
1 = Yes _____, 2 = No _____, 8 = DK _____, 9 = NR _____
If so, what is the sequence?
(3), Are these places connected by a trail?
1 = Yes _____, 2 = No _____, 8 = Don't Know _____, 9 = No Response _____
(4), If these trails are connected by a trail, would you call such a trail a "pilgrimage trail"?
1 = Yes _____, 2 = No _____, 8 = DK_____, 9 = NR______

Or would you suggest another name for the trail?

(5), If another name, what would you call the trail? _______________________________

(6), Please draw a map to show how these local places are connected and if they are connected in terms of a ceremonial sequence please number the places by that sequence.

**Regional Cultural Landscapes**

(7), Are these ceremonial places connected with other places in California, Nevada, Arizona, and Utah?

1 = Yes ______, 2 = No ______, 8 = Don't Know ______, 9 = No Response ______

If so, which ones are connected.

Place _________ to place _____________; connected physically_____ or spiritually ______ or both.

Place _________ to place _____________; connected physically_____ or spiritually ______ or both.

Place _________ to place _____________; connected physically_____ or spiritually ______ or both.

Place _________ to place _____________; connected physically_____ or spiritually ______ or both.

Place _________ to place _____________; connected physically_____ or spiritually ______ or both.

Place _________ to place _____________; connected physically_____ or spiritually ______ or both.

Place _________ to place _____________; connected physically_____ or spiritually ______ or both.

(8), If these places are connected, were they used in sequence?

1 = Yes ______, 2 = No ______, 8 = DK ______, 9 = NR ______

(9), If so, what is the sequence?

(10), Are these places connected by a trail?

1 = Yes ______, 2 = No ______, 8 = Don't Know ______, 9 = No Response ______

**(11), Please draw a map to show how they are connected and if they are connected in terms of a ceremonial sequence please number the places by that sequence.**

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
