# Peer review of "Sustainable Heritage Tourism: Native American Preservation Recommendations at Arches, Canyonlands, and Hovenweep National Parks"

_sustainability, doi:10.3390/su12239846_

Round 1

Reviewer 1 Report

The suggestions have been attended by autrhors

Author Response

Reviewer # 1:

Open Review

English language and style

( ) Extensive editing of English language and style required
(x) Moderate English changes required
( ) English language and style are fine/minor spell check required
( ) I don't feel qualified to judge about the English language and style

Yes

Can be improved

Must be improved

Not applicable

Does the introduction provide sufficient background and include all relevant references?

(x)

( )

( )

( )

Is the research design appropriate?

(x)

( )

( )

( )

Are the methods adequately described?

(x)

( )

( )

( )

Are the results clearly presented?

(x)

( )

( )

( )

Are the conclusions supported by the results?

(x)

( )

( )

( )

Comments and Suggestions for Authors

The suggestions have been attended by autrhors

Reviewer 2 Report

Dear authors:

Thank you very much for your paper.

The creation of new cultural heritage in response to the demands of the local population is an issue that shows great respect for society. In addition, tourism is currently a fundamental eco-economic activity and it is necessary to design strategies to make it more sustainable. Therefor, I consider your paper is at the center of scientific debates about tourism sustainability.
Thus, I would like to make a series of suggestions for improvement the paper: 

1. Review of formal aspects, specially, foodnotes and bibligoraphy (for example, lines 51, 58 and 70).

2. The bibliographic review should be reinforced. Expand and analyze the number of publications that develop similar themes. 

3.The theoretical framework must be clearly indicated, it is understood that it is sustainable development, but it is based on different pillars (social, economic, environmental, cultural). I suggest that the theoretical framework be linked to cultural sustainability, specifically, to the role that cultural heritage plays in sustainable developement. I hope you find the following articles useful: Soini and Bikerland (2012) and Soini and Dessein (2014).

4. It must clearly specify the working hypothesis and the objectives of the investigation.

5. You should explain the methodology in more detail. Incorporate an example of the interview model.

6. Table 1 shows the scope of work. It has hardly been exploited. I recommend that the information in the table be expanded.

7.The discussion should be reoriented, at least the debate on the impact of the tourist. Actually, the article does not study the impact of tourism, rather how a correctly interpreted heritage can help tourists to be more respectful with the cultural heritage and the lanscape. I suggest redirecting that part of the discussion to that topic. 

8. Future research lines should be indicated with more detail.

On the other hand, the research raises a scientific question to me. In reality, the recommendations indicated in the three case studies have not been implemented. So we don't know how they affect the development of sustainable tourism in the area. From a scientific point of view, this study does not show whether these cultural resources generate sustainability. For this reason, I suggest reviewing the story line of the article and propose new approaches.

I hope these guidelines help you to improve the paper. 

Regards

Author Response

Reviewer #2:

Thank you very much for your paper.

The creation of new cultural heritage in response to the demands of the local population is an issue that shows great respect for society. In addition, tourism is currently a fundamental eco-economic activity and it is necessary to design strategies to make it more sustainable. Therefor, I consider your paper is at the center of scientific debates about tourism sustainability. Thus, I would like to make a series of suggestions for improvement the paper: 

1. Review of formal aspects, specially, foodnotes and bibligoraphy (for example, lines 51, 58 and 70).

Author Response: These have been corrected

  1. The bibliographic review should be reinforced. Expand and analyze the number of publications that develop similar themes. 

Author Response: this has been done

3.The theoretical framework must be clearly indicated, it is understood that it is sustainable development, but it is based on different pillars (social, economic, environmental, cultural). I suggest that the theoretical framework be linked to cultural sustainability, specifically, to the role that cultural heritage plays in sustainable developement. I hope you find the following articles useful: Soini and Bikerland (2012) and Soini and Dessein (2014).

Author response: Thanks for the new references, some have been added

  1. It must clearly specify the working hypothesis and the objectives of the investigation.

Author response: this has been expanded

  1. You should explain the methodology in more detail. Incorporate an example of the interview model.

Author response: This has occurred and a new Appendix is added

  1. Table 1 shows the scope of work. It has hardly been exploited. I recommend that the information in the table be expanded.

Author response: this has been done

7.The discussion should be reoriented, at least the debate on the impact of the tourist. Actually, the article does not study the impact of tourism, rather how a correctly interpreted heritage can help tourists to be more respectful with the cultural heritage and the lanscape. I suggest redirecting that part of the discussion to that topic. 

Author response: lot of new text in both areas

  1. Future research lines should be indicated with more detail.

On the other hand, the research raises a scientific question to me. In reality, the recommendations indicated in the three case studies have not been implemented. So we don't know how they affect the development of sustainable tourism in the area. From a scientific point of view, this study does not show whether these cultural resources generate sustainability. For this reason, I suggest reviewing the story line of the article and propose new approaches.

Author response: many additions regarding this suggestion.

I hope these guidelines help you to improve the paper. 

Regards

Author response: thanks for recommendations. They were useful.

Reviewer 3 Report

I would appreciate the authors’ research on Sustainable Heritage Tourism. However, to this reviewer, the manuscript needs further revisions to be considered for publishing due to its methodological flaws. Below, please find my main concerns for such an opinion.

First, the research background needs to provide a broader vision of tourism challenges in the cultural heritage sites. Tourism educations programs have been in practice in many European countries, but they have not achieved their idealistic goals. For instance, please see [Tourism in the New Europe: The Challenges and Opportunities of EU Enlargement], [https://doi.org/10.3390/su9101879], and [https://doi.org/10.3390/su12093929]. These studies explain the practice and challenges of sustainable heritage tourism. Explaining the tourism challenges and mentioning the best practices are essential for expanding an up to date discussion.

Second, in terms of methodology, there is a critical issue in the applied approach. The conceptualization of cultural heritage is vague in this study, and different concepts are considered equal. The authors refer to the term “Cultural Landscape” without clearly define its definition (line 71, what is it? And how to contribute?), and inline 58, they have considered it as “heritage places.” A cultural landscape is a heritage resource that reveals a cultural community evolved interactions with the natural landscape. It is a part of the territory “as perceived by people, whose character is the result of the action and interaction of natural and human factors.” When we address the cultural landscape, we need to understand its link with the territorial dynamics and sustainable tourism management plans in these resources. Due to the weakness in defining the concept, these dimensions are not adequately considered in the article.

Third, there are some fundamental questions in this study that the discussion section needs to answer them. Please explain more in detail what knowledge, abilities, skills, and attitude are essential for tourism education in Arches National Park, Canyonlands National Park, and Hovenweep National Monument. How can tourist education be assessed in terms of mass tourism? Furthermore, considering the cultural landscape's proposed definition, how can regional planning measures contribute to such a practice?

Fourth, I wonder why the paper is not written in the template of Sustainability and has ended without a conclusion section.

Author Response

Reviewer #3:

Top of Form

I would appreciate the authors’ research on Sustainable Heritage Tourism. However, to this reviewer, the manuscript needs further revisions to be considered for publishing due to its methodological flaws. Below, please find my main concerns for such an opinion.

First, the research background needs to provide a broader vision of tourism challenges in the cultural heritage sites. Tourism educations programs have been in practice in many European countries, but they have not achieved their idealistic goals. For instance, please see [Tourism in the New Europe: The Challenges and Opportunities of EU Enlargement], [https://doi.org/10.3390/su9101879], and [https://doi.org/10.3390/su12093929]. These studies explain the practice and challenges of sustainable heritage tourism. Explaining the tourism challenges and mentioning the best practices are essential for expanding an up to date discussion.

Author response: Thanks for the references. Useful. Many changes have occurred.

Second, in terms of methodology, there is a critical issue in the applied approach. The conceptualization of cultural heritage is vague in this study, and different concepts are considered equal. The authors refer to the term “Cultural Landscape” without clearly define its definition (line 71, what is it? And how to contribute?), and inline 58, they have considered it as “heritage places.” A cultural landscape is a heritage resource that reveals a cultural community evolved interactions with the natural landscape. It is a part of the territory “as perceived by people, whose character is the result of the action and interaction of natural and human factors.” When we address the cultural landscape, we need to understand its link with the territorial dynamics and sustainable tourism management plans in these resources. Due to the weakness in defining the concept, these dimensions are not adequately considered in the article.

Author response: many additions to the methodology including a new appendix to extend the discussion.

Third, there are some fundamental questions in this study that the discussion section needs to answer them. Please explain more in detail what knowledge, abilities, skills, and attitude are essential for tourism education in Arches National Park, Canyonlands National Park, and Hovenweep National Monument. How can tourist education be assessed in terms of mass tourism? Furthermore, considering the cultural landscape's proposed definition, how can regional planning measures contribute to such a practice?

Author response: Native representatives worked with NPS staff to explain and educate tourists.

Author response: cultural landscape protection is a problem produced by administrative differenced. The hoodoo landscape begins in a national park, extends across a river controlled by the Bureau of Recreation, go up slope toward the mountains across lands controlled by the Bureau of Land Management, and ends in the La Sal Sky Island managed by the Forest Service. Despite efforts to create cross agency management of culgtural resources like landscapes, there are few cases where this has occurred with a cultural landscape.

Fourth, I wonder why the paper is not written in the template of Sustainability and has ended without a conclusion section.

Author response: This problem has been improved.

Author response: Thanks for the careful reading of the paper and good comments.

Reviewer 4 Report

To begin with, I wanted to thank the authors for bringing their readers along this journey. I am glad to see more discussion of Native interpretations and relationships with the land. It can only help archaeologists such as myself to better understand the impact of the work we do. This is a valuable contribution to the literature.

That said, I do have some general comments. Specific comments are provided in the marked up PDF, which is attached to this review. 

First, there's a lot of necessary, albeit minor, grammatical corrections needed. Rather than reproduce them all here, please refer to the marked up PDF. I've highlighted areas that need editing (spellcheck, capitalization, etc). The errors don't take away from the authors' argument, though it does unfortunately come across as unprofessional. 

Second, there are many statements made throughout the manuscript that would benefit from references/citations to appropriate literature. I have highlighted these within the PDF. 

Third, the manuscript ends very abruptly. It would benefit greatly from a short conclusion that applies the lessons learned and recommendations for Southwest US parks to the greater concept of sustainable heritage tourism. The authors begin that discussion in their section on "Sustainable Tourism Development" but I would like them to continue that discussion a bit. 3 to 4 paragraphs would likely do, even if it's just to highlight how the authors see that this research can be the foundation for future research or changes in the practice of park management.

Finally, I want to repeat my thanks. It's an honor to have Native voices share these stories and viewpoints with the larger, colonist society (especially so with a settler archaeologist). This is an important conversation and I hope to see this particular contribution to that conversation in print soon. I know I would reference it in my own work. 

Author Response

Reviewer #4:

Comments and Suggestions for Authors

To begin with, I wanted to thank the authors for bringing their readers along this journey. I am glad to see more discussion of Native interpretations and relationships with the land. It can only help archaeologists such as myself to better understand the impact of the work we do. This is a valuable contribution to the literature.

Author response: Want to thank this reviewer for the useful and insightful comments. This reviewer clearly understood the primary intentions of the manuscript and helped achieve those goals. The track changes in the pdf were wonderful.

That said, I do have some general comments. Specific comments are provided in the marked up PDF, which is attached to this review. 

First, there's a lot of necessary, albeit minor, grammatical corrections needed. Rather than reproduce them all here, please refer to the marked up PDF. I've highlighted areas that need editing (spellcheck, capitalization, etc). The errors don't take away from the authors' argument, though it does unfortunately come across as unprofessional.

Author response: We worked to resolve these, thanks for catching them. 

Second, there are many statements made throughout the manuscript that would benefit from references/citations to appropriate literature. I have highlighted these within the PDF. 

Author response: thanks for the references. We used many of them.

Third, the manuscript ends very abruptly. It would benefit greatly from a short conclusion that applies the lessons learned and recommendations for Southwest US parks to the greater concept of sustainable heritage tourism. The authors begin that discussion in their section on "Sustainable Tourism Development" but I would like them to continue that discussion a bit. 3 to 4 paragraphs would likely do, even if it's just to highlight how the authors see that this research can be the foundation for future research or changes in the practice of park management.

Author response: we have tried to strengthen these.

Finally, I want to repeat my thanks. It's an honor to have Native voices share these stories and viewpoints with the larger, colonist society (especially so with a settler archaeologist). This is an important conversation and I hope to see this particular contribution to that conversation in print soon. I know I would reference it in my own work. 

Author response: We thank this reviewer for the help and time spent reading the manuscript.

Round 2

Reviewer 2 Report

Thanks very much for the improvements.

Regards

Reviewer 3 Report

The revised version has satisfactorily responded to the main concerns proposed by this reviewer. I would suggest it for publication in its current format. Thanks to the authors, they made a nice improvement. 

This manuscript is a resubmission of an earlier submission. The following is a list of the peer review reports and author responses from that submission.

Round 1

Reviewer 1 Report

Th figures - photos - should have copyright symbol and the name of the Author.

There is no photo credits or official authorization about this subject.

 We undersantd that t is research material, but photos are photos wirht credits nowadays.

This is a minor review to be accpetd

Reviewer 2 Report

Dear authors:

First of all, I would like to thank you for working in such an interesting field as the enhancement of the cultural heritage of native American peoples is. It is necessary to continue researching in this field to reinforce protection and conservation of the natives american peoples' cultural heritage and give it the importance it deserves.

However, I consider that the publication of the article is not possible for the following reasons:

  • I think there is a repetition in case studies 1 and 2. The same contents are repeated in lines 319-386 and 411-468. Therefore, the conclusions are not too viable.
  • The main research question as well as the working hypotheses and objectives should be clearly stated.
  • I consider that the state of the art that can't be considered academic enough. Studies already published with the same topic should be pointed out, clearly indicating the results.
  • The methodology should develop further. It should be clearly indicated in the article for example, how many people participated in the study in each case study, when, where, how the data were recorded, etc.

I would also like to make a number of suggestions about the discussion:

  • Taking into account the nature of the experiments developed, focused on the valuation of new cultural elements as cultural heritage of the Native Americans, I suggest focusing the discussion on the expansion of cultural heritage as the axis of sustainable tourism. 
  • The impacts of tourists on cultural heritage have not been addressed in the experiments, nor has economic development. The relationship of these two themes to the main topic seems a bit forced. A more coherent line of discussion should be found.

I thank the authors again for their work. I emphasize again the importance of studies like the one you have presented. I hope the article can be published in the future.

Reviewer 3 Report

The manuscript entitled “Sustainable Heritage Tourism: Native American Preservation Recommendations at Arches, Canyonlands, and Hovenweep National Parks” aims to understand the potential impacts of heritage tourism to the selected Native American places at Arches National Park, Canyonlands National Park, and Hovenweep National Monument. To this reviewer, the manuscript has serious flaws both in terms of methodology and the structure. Below, please find my general concerns for such an opinion.
First: The state of the art is not clearly described in the abstract/introduction sections. It seems that the authors have considered Heritage Tourism as a solution for the conservation of Native American communities and their built environment. However, it should be mentioned that most of the UNESCO world heritage sites have been under pressure of mass tourism during the last years. So, when tourism is usually seen as a conservation strategy, most management plans become unable to face mass-tourism, which threatens the local community's everyday life. I would suggest the authors read one of the most recent paper in this respect at [https://doi.org/10.3390/su12093929].
Second:
There is no clear definition of the used terms, and in most cases, these terms are considered equivalent to each other. For instance, what is “native heritage”? Do you mean “cultural heritage site”? If so, what do you mean by “cultural landscape”? For historical conservation, these terms have different approaches for the assessment of historical value. The lack of terminology is evident in the manuscript.
Third:
The research methodology is ambiguous, or at least not well-described. It is unclear how the questionnaire was designed? How has the credibility of the answers endorsed? Considering the nature of the paper, it is important to note the applied ethical principles in this paper.
Fourth:
Why has the conclusion section missed? What are the real findings of this article? How it contributes to the discourse in this field?